# BEYOND MARKOVIAN: REFLECTIVE EXPLORATION VIA BAYES-ADAPTIVE RL FOR LLM REASONING

**Shenao Zhang**[1*]**, Yaqing Wang**[2]**, Yinxiao Liu**[2]**, Tianqi Liu**[2]**, Peter Grabowski**[3]**, Eugene Ie**[3]**,
Zhaoran Wang**[1†]**, Yunxuan Li**[3†]
[1]Northwestern University, [2]Google DeepMind, [3]Google

## ABSTRACT

Large Language Models (LLMs) trained via Reinforcement Learning (RL) have exhibited strong reasoning capabilities and emergent reflective behaviors, such as rethinking and error correction, as a form of in-context exploration. However, the Markovian policy obtained from conventional RL training does not give rise to reflective exploration behaviors since the policy depends on the history only through the state and therefore has no incentive to enrich identical states with additional context. Instead, RL exploration is only useful during training to learn the optimal policy in a trial-and-error manner. Therefore, it remains unclear whether reflective reasoning will emerge during RL, or why it is beneficial. To remedy this, we recast reflective exploration within a Bayesian RL framework, which optimizes the expected return under a posterior distribution over Markov decision processes induced by the training data. This Bayesian formulation admits uncertainty-adaptive policies that, through belief updates, naturally incentivize information-gathering actions and induce self-reflection behaviors. Our resulting algorithm, BARL, instructs the LLM to stitch and switch strategies based on the observed outcomes, offering principled guidance on when and how the model should reflectively explore. Empirical results on both synthetic and mathematical reasoning tasks demonstrate that BARL outperforms conventional RL approaches, achieving superior test-time performance and token efficiency. Our code is available at https://github.com/shenao-zhang/BARL.

## 1 INTRODUCTION

Large Language Models (LLMs) have demonstrated impressive reasoning abilities, such as in solving complex math problems. A key factor driving this progress is the use of Chain-of-Thought (CoT) reasoning (Wei et al., 2022), where the model engages in intermediate deliberation before producing an answer. Building on this, recent advances have employed Reinforcement Learning (RL) to further enhance LLM reasoning by optimizing for verifiable outcome rewards (Jaech et al., 2024; Guo et al., 2025; Yu et al., 2025; Wei et al., 2025). Notably, RL-trained models have exhibited emergent behaviors such as engaging in self-reflection, a process of revisiting previous states to correct earlier mistakes (Guo et al., 2025; Zeng et al., 2025), which offers a potential way for test-time scaling (Snell et al., 2024; Brown et al., 2024) by generating longer CoTs. However, despite these compelling phenomena, it remains unclear why and under what conditions self-reflections are beneficial, or whether such behaviors will emerge through conventional RL training.

In this work, we use reflective exploration to refer to behavior where the model revisits a previously visited state and, informed by the additional context it has acquired, alters its subsequent actions. This definition links self-reflective directly to exploration under uncertainty, providing a principled basis for when and why strategy switches should occur rather than treating them as ad hoc quirks of the model's CoTs. We formalize this via strictly uncertainty-adaptive policies whose actions depend nontrivially on the belief. Since conventional RL typically admits a Markovian optimal policy, reflective exploration is not induced since the policy depends on the history only through the state and is thus not incentivized to enrich identical states with additional context. Instead, RL explores during training through repeated trial and error, with no incentives for further exploration when

---

*Work done during an internship at Google. †Equal advising.

parameters are frozen after deployment. Therefore, under conventional RL, there is no guarantee that self-reflections will emerge during training, nor does it explain why reflective explorations might be advantageous when the trained policy is deployed at test time.

To address this gap, we propose grounding reflective reasoning with Bayesian RL, which maximizes the expected return in the deployment stage under a posterior distribution over Markov Decision Processes (MDPs) induced by the training data. The objective incentivizes both reward-seeking actions and exploration actions that gather information to reduce the MDP's uncertainty, such as the reward uncertainty regarding the progress made by different actions. We show that optimizing this Bayesian RL objective yields uncertainty-adaptive policies that adapt on-the-fly by updating their beliefs and switching strategies based on observed outcomes, naturally leading to reflective exploration behaviors. Moreover, we prove that the optimal adaptive policy can achieve arbitrarily higher Bayes-expected return than any optimal Markovian policy.

Building upon this formulation, we introduce a novel algorithm, *Bayes-Adaptive RL for LLM Reasoning* (BARL). For each prompt, BARL performs online rollouts to generate a set of candidate answers, each associated with an MDP hypothesis. The state-action value is then computed by weighting each hypothesis according to the model's current belief, with penalties applied for mismatches between predicted and observed rewards, thereby signaling when to switch strategies. BARL provides a principled mechanism for integrating and revising plausible strategies, analogous to linearizing best-of-N, but with explicit guidance on *when* and *how* the model should reflectively explore.

To illustrate the benefits of BARL, we begin with a synthetic task designed to mirror the training-deployment procedure in LLM reasoning. The agent receives a reward only when it repeats a prompt token exactly three times, but the training and evaluation prompt tokens differ. Conventional RL fails to generalize beyond the training prompt tokens. In contrast, BARL learns to switch strategies by eliminating hypotheses, ultimately discovering the ground-truth MDP for optimal behavior.

We further evaluate BARL on math reasoning tasks using various LLMs, including Qwen2.5-Math-1.5B, Qwen2.5-Math-7B, and R1-Distill-Llama-8B. Across these models, BARL consistently outperforms conventional RL algorithms, such as GRPO and a strong progress-reward baseline, on multiple benchmarks. BARL achieves significantly greater token efficiency, requiring up to **1.63x** fewer average tokens than the progress baseline, **2x** fewer than GRPO, and over **10x** fewer than the Qwen2.5-Math-1.5B base model. Moreover, we observe no strong correlation between overall model performance and the frequency of reflections. Instead, BARL's advantage stems from more efficient exploration and more effective thinking tokens. We summarize the key takeaways below:

---

**Key Takeaways: Why, How, and When Should LLMs Reflectively Explore**

- **Why:** Conventional RL neither ensures the emergence of reflective exploration nor explains its benefits since (1) RL typically attains Markovian optimal policies, which, cannot exhibit reflective exploration that collects additional context and act differently when revisiting the same state, and (2) RL exploration is confined to trial-and-error during training, leaving no mechanism that encourages adaptive, belief-driven exploration at deployment. In contrast, Bayesian RL, by optimizing Bayes-expected return during deployment, encourages exploration to gather contextual information that reduces the MDP uncertainty.

- **How:** BARL provides a principled way to stitch plausible strategies by maintaining a posterior over MDP hypotheses, each associated with a sampled candidate answer. Reflective exploration emerges through hypothesis elimination, enabling on-the-fly adaptation.

- **When:** LLMs should self-reflect when discrepancies arise between their internal beliefs and cumulative reward feedback—signaling strategy switching by downweighting hypotheses that have high belief probabilities but are unlikely to be optimal given previous observations.

---

## 2 RELATED WORK

**LLM Reasoning.** As an emerging capability of model scale, LLMs can generate intermediate CoTs to solve complex reasoning tasks (Wei et al., 2022; Kojima et al., 2022) and scale test-time performance by allocating more thinking tokens (Snell et al., 2024; Brown et al., 2024). Early efforts enhanced LLM reasoning via supervised fine-tuning on human-annotated data (Cobbe et al.,

2021; Yue et al., 2023; Yu et al., 2023) or linearized search traces (Lehnert et al., 2024; Gandhi et al., 2024). However, due to the distribution shift between LLM responses and curated data, LLM-generated data has proven effective through rejection sampling (Dong et al., 2023; Yuan et al., 2023) by filtering out low-quality rationales (Zelikman et al., 2022; 2024) or with EM iterations (Singh et al., 2023; Zhong et al., 2025; Zhang et al., 2025). Recently, RL has gained increasing interest for improving reasoning (Aksitov et al., 2023; Havrilla et al., 2024; Wang et al., 2024a; Shao et al., 2024). Process rewards (Uesato et al., 2022; Lightman et al., 2023) with Monte Carlo unrolls (Kazemnejad et al., 2024; Wang et al., 2023a; 2024b; Luo et al., 2024) offer finer-grained feedback but are computationally expensive. Outcome-reward RL (Guo et al., 2025) demonstrates emergent deliberative reasoning abilities such as self-reflection. Yet, limited work has investigated the underlying mechanisms of such behaviors. In fact, recent findings suggest that reflections do not consistently emerge from RL training and exhibit weak correlation with performance (Liu et al., 2025). Similar to our work, Xiang et al. (2025); Qu et al. (2025) also study reflective exploration of LLMs, from a meta-RL (Duan et al., 2016) perspective: Xiang et al. (2025) justifies deliberative reasoning as providing extra CoT state contexts, and Qu et al. (2025) uses progress reward (Setlur et al., 2024) to reduce regret in outcome-reward RL. Our method differs from Xiang et al. (2025) in that we ground reflective reasoning in environment rewards, rather than relying solely on the internal CoT states generated by the model itself. Compared to Qu et al. (2025), which rewards strategies that make progress towards the correct answer, BARL additionally encourages exploring plausible strategies under the Bayesian framework. We defer a more detailed discussion to the end of Section 4. We experimentally compare with a variant of Qu et al. (2025) that estimates progress using answer probability differences. Prior work has also brought Bayesian exploration principles to LLMs: Arumugam & Griffiths (2025) implements *Posterior Sampling for Reinforcement Learning* (PSRL), a statistically-efficient algorithm, using LLMs as modular components, while Dwaracherla et al. (2024) shows that active query selection with epistemic neural networks and double Thompson sampling reduces the amount of human feedback required to train reward models for RLHF. In contrast, we optimize a Bayesian objective directly during RL fine-tuning of the LLM policy, rather than at the level of external algorithmic scaffolding or data collection, to provide a step-level guidance on when and how LLMs should self-reflect. Besides, our method differs from Wang et al. (2023b); Qiu et al. (2023) that manually designs hypothesis proposal–selection pipelines.

**Reinforcement Learning.** Conventional RL explores only during training. Exceptions include works that explicitly maximum entropy objectives (Haarnoja et al., 2018) to learn stochastic policies, primarily to accelerate *training* convergence in settings where evaluation remains in-distribution, such as robotic control. Bayes-Adaptive RL (Bellman & Kalaba, 1959; Duff, 2002; Guez et al., 2012; Ghavamzadeh et al., 2015; Lidayan et al., 2025; Liu et al., 2023) has been studied to pursue the optimal exploration-exploitation trade-off in uncertain environments. When the true MDP identity is latent and must be inferred from interaction (states, actions, rewards), Bayesian RL connects naturally to Partially Observable MDPs (Duff, 2001; Ghosh et al., 2021). Exact solutions of the Bayesian RL objective are often intractable, promoting the development of approximate methods (Guez et al., 2012; Zhang, 2022; Chen et al., 2024). In our work, we adopt a policy gradient that operates over candidate answers, which differs from Ghosh et al. (2022) that leverages value ensembles in offline RL and Qiu et al. (2025) that applies SFT on an oracle Bayesian model's outputs.

## 3  PROBLEM FORMULATION

**LLM Reasoning.** A finite-horizon MDP $\mathcal{M}$ can be defined by the state space $\mathcal{S}$, action space $\mathcal{A}$, horizon $T$, and reward function $r_{\mathcal{M}}(s, a)$, where $s \in \mathcal{S}$ and $a \in \mathcal{A}$. Here, the initial state $s_0$ is the prompt, and the action $a_t$ is the $t$-th step of the CoT, which can be either separated by special tokens (Wang et al., 2023a) or defined as a fixed length of reasoning tokens (Luo et al., 2024). We adopt the latter definition due to its simplicity. The state transition is deterministic by appending the new reasoning step, i.e., $s_{t+1} = s_t + a_t$.

We are interested in a training-deployment procedure. During training, the agent only receives information about the true MDP $\mathcal{M}^* := (\mathcal{S}, \mathcal{A}, r, T)$ via the input training data $\mathcal{D}$. During deployment, we freeze the learned policy and deploy it in $\mathcal{M}^*$. Notably, $\mathcal{D}$ does not uniquely identify $\mathcal{M}^*$, inducing epistemic uncertainty about the MDP. Formally, $\mathcal{D}$ and a prior distribution over MDPs $p(\mathcal{M})$ define a posterior distribution over MDPs, $p(\mathcal{M}|\mathcal{D}) \propto p(\mathcal{M})p(\mathcal{D}|\mathcal{M})$. The objective for policy $\pi$ is

thus to maximize return in expectation over MDPs from the posterior $p(\mathcal{M}|\mathcal{D})$:

$$\mathcal{J}(\pi) = \mathbb{E}_{\mathcal{M} \sim p(\mathcal{M}|\mathcal{D})}\Big[\mathcal{J}_{\mathcal{M}}(\pi)\Big], \quad \text{where } \mathcal{J}_{\mathcal{M}}(\pi) = \mathbb{E}_{s_0, \pi}\left[\sum_{t=0}^{T-1} r_{\mathcal{M}}(s_t, a_t)\right]. \tag{3.1}$$

This $\mathcal{J}(\pi)$ objective is designed to, for any data $\mathcal{D}$, train a policy $\pi$ that maximizes its Bayes-expected return in the unknown true environment during deployment, i.e., its performance averaged over MDPs drawn from the posterior $p(\mathcal{M}|\mathcal{D})$.

**Progress Reward.** Prior work (Uesato et al., 2022; Guo et al., 2025) employs an outcome-level reward verifier$(s_T, y_{s_0}^*)$ for the true MDP $\mathcal{M}^*$, which uses a verifier to perform a regular expression match (either 0 or 1) between $s_T$ and the ground-truth answer $y_{s_0}^*$ corresponding to the prompt $s_0$. We extend this sparse-reward setting by incorporating a *progress* reward (Setlur et al., 2024; Qu et al., 2025), which quantifies the increase of the model's probability of outputting $y_{s_0}^*$ after appending $a$ at the CoT $s$, i.e., for $0 \le t \le T-1$:

$$r(s_t, a_t) = \pi_\phi(y_{s_0}^* \mid s_t + a_t + \texttt{</think>}) - \pi_\phi(y_{s_0}^* \mid s_t + \texttt{</think>}), \tag{3.2}$$

where $\texttt{</think>}$ is the end sign of thinking, such as the answer elicitation prompt "Based on the above reasoning, the answer is $\texttt{\textbackslash boxed}$" that we adopt, and $\pi_\phi$ is any policy. Compared to Monte-Carlo process rewards (Luo et al., 2024; Qu et al., 2025), (3.2) is computationally efficient by avoiding multiple branched rollouts at each step, and the KV cache of $s_{0:T}$ during rollouts can also be reused. The above definition naturally extends to any $r_{\mathcal{M}}$ defined w.r.t. the answer $y_{s_0}^{\mathcal{M}}$. The Q-value is then

$$Q_{\mathcal{M}}^\pi(h_t, a_t) = \mathbb{E}_\pi\left[\sum_{t'=t}^{T-1} r_{\mathcal{M}}(s_{t'}, a_{t'}) + \text{verifier}(s_T, y_{s_0}^*)\right], \tag{3.3}$$

where $h_t = (s_0, a_0, r_0, \ldots, s_t)$ is the history and we define $s(h_t) = s_t$ as the final state $s_t$ from $h_t$.

**Reflective Exploration.** Informally, we define reflective exploration as the pattern in which the LLM revisits a prior state after an in-context exploratory step to take different actions at that state. Specifically, a language reflection step such as "Let's reconsider the geometric relationship" corresponds to re-visiting and retrying. We illustrate this using a binary search tree as in the right figure: for the trajectory $s_0 \to s_1 \to s_2 \to s_1 \to s_3$, $s_1 \to s_2$ is an exploration step, $s_2 \to s_1$ is a reflective step that signals the strategy switch from $s_2$ to $s_3$, and the "geometric relationship" in the above example originates from $s_1$. We formalize this behavior through the lens of adaptive policies.

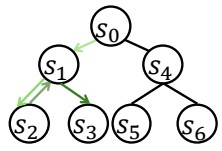

Figure 1: An example of reflective reasoning trajectories.

**Definition 3.1** (Uncertainty-Adaptive Policy)**.** Define the belief as the posterior over MDPs $b(h_t)(\mathcal{M}) \coloneqq p(\mathcal{M} \mid h_t, \mathcal{D})$. A policy $\pi$ is *uncertainty-adaptive* if there exists a measurable mapping $\mu : \mathcal{S} \times \mathcal{P}(\mathcal{M}) \to \Delta(\mathcal{A})$ such that for all $h_t$ and $a_t$, $\pi(a_t \mid h_t) = \mu(a_t \mid s(h_t), b(h_t))$. If in addition, there exists $h_t$ and $h_t'$ such that $s(h_t) = s(h_t')$ but $\mu(a_t \mid s(h_t), b(h_t)) \ne \mu(a_t \mid s(h_t'), b(h_t'))$, then $\pi$ is *strictly* uncertainty-adaptive, or equivalently, not Markovian.

From this perspective, reflective exploration is the behavioral signature of an uncertainty-adaptive policy: as the agent's belief evolves, it rationally revisits previous states and generates a different action distribution. In contrast, a purely state-only policy that ignores this evolving belief would respond identically upon revisiting the same state. Formally,

**Definition 3.2** (Reflective Exploration)**.** We say that a policy $\pi$ exhibits *reflective exploration* if there exist $t_1 > t_2$ and $h_{t_1}, h_{t_2}$ such that

$$\varphi(s(h_{t_1})) = \varphi(s(h_{t_2})), \qquad \pi(\cdot \mid h_{t_1}) \ne \pi(\cdot \mid h_{t_2}),$$

where $\varphi : \mathcal{S} \to \mathcal{Z}$ is a measurable mapping for which there exists $\vartheta : \mathcal{Z} \times \mathcal{H}_{\mathcal{R}} \to \Delta(\mathcal{A})$ satisfying for every $h_t$ that (1) $\vartheta(\cdot \mid \varphi(s_t), r_{0:t-1}) = \pi(\cdot \mid h_t)$; and (2) if there is some process $X_t$ and some $\upsilon$ such that $\pi(\cdot \mid h_t) = \upsilon(\cdot \mid X_t)$ then $X_t = \psi(\varphi(s_t))$ for some measurable $\psi$.

Intuitively, it describes an agent that returns to the same underlying latent state but, in light of what it has learned from past rewards, deliberately chooses a different strategy than it did before. The latent state representation $\varphi$ abstracts away superficial differences in the text state. The existence of $\vartheta$ and $\upsilon, \psi$ formalizes the latent factorization and state-minimality properties of $\varphi$, respectively.

# 4 THE NECESSITY OF BAYESIAN RL FOR REFLECTIVE EXPLORATION

**Conventional RL.** The RL objective for policy $\pi$ is $\mathcal{J}_{\mathcal{M}^*}(\pi)$ in (3.1). It is widely known that the optimal policy and value satisfy the Markov property by depending on $h_t$ only through the state $s_t$.

**Theorem 4.1** (RL Admits Markovian Optimal Policy). For any MDP, there exists a Markovian policy $\pi^*$, i.e., a policy that is not strictly uncertainty-adaptive, that maximizes the RL objective. Every policy that exhibits reflective exploration is not Markovian.

The result indicates that the RL objective admits Markovian policies to be optimal. Optimizing a Markovian policy, however, does not induce reflective exploration behaviors. Intuitively, action sequences whose effect is only to enrich the history $h_t$ (e.g., incorrect attempts followed by revisiting the same state) do not change the sufficient state representation $s_t$ and therefore cannot improve the value of an optimal Markovian policy.

Instead, in RL, exploration is only useful during training, in a trial-and-error manner with repeated episodes, to discover a return-maximizing action sequence. The learned policy is fully exploited at deployment time with no incentive for further exploration since no policy updates occur, which is why $\epsilon$-greedy often sets $\epsilon \approx 0$ when deployed (Mnih et al., 2015; Hessel et al., 2018).

**Bayesian RL.** In Bayesian RL, the agent maintains uncertainty over the underlying MDP, which is gradually reduced through interactions. Due to this epistemic partial observability (Duff, 2001; Ghosh et al., 2021), the policy and value depend on the full history $h_t$, instead of only the state $s_t$, to capture the agent's evolving belief about the MDP parameters through cumulative observations. The objective encourages the agent to not only maximize immediate rewards based on the current belief but also to explore to gather more context about the uncertain MDP.

**Theorem 4.2** (Bayesian RL Admits Adaptive Optimal Policy). For the Bayesian RL objective in (3.1) associated with any $(\mathcal{D}, p(\mathcal{M}))$, there exists an optimal policy $\pi^*$ that is uncertainty-adaptive.

The above result shows that optimizing the Bayesian RL objective in (3.1) attains an uncertainty-adaptive policy and thus naturally inducing reflective exploration. While reflective actions may be suboptimal relative to the (unknown) ground-truth MDP, the gathered context, especially rewards, reduces the MDP uncertainty. Future policies can thus leverage the updated belief to act optimally.

Theorem 4.2 should be read as a structural result: for any $(\mathcal{D}, p(\mathcal{M}))$, one can always represent a Bayes-optimal policy to be uncertainty-adaptive. It is possible that the dependence on the belief can be vacuous and the policy degenerates to Markovian. However, we will show in the following that there exist instances where any Bayes-optimal policy must be strictly uncertainty-adaptive, and the best Markovian policy can far underperform the optimal adaptive policy.

**Theorem 4.3** (Gap Between Markovian and Adaptive Policies). There exist instances $(\mathcal{D}, p(\mathcal{M}))$ for which the objective in (3.1) is maximized by a strictly uncertainty-adaptive policy. Moreover, the performance gap between Markovian and strictly adaptive policies can be arbitrarily large.

*Sketch of proof.* Consider the binary decision structure in Fig. 1. Let the prior $p(\mathcal{M}_1) = p(\mathcal{M}_2) = p(\mathcal{M}_3) = p(\mathcal{M}_4) = 1/4$, where $r_{\mathcal{M}_1}(s_2) = r_{\mathcal{M}_2}(s_3) = r_{\mathcal{M}_3}(s_5) = r_{\mathcal{M}_4}(s_6) = 1$ all other rewards are zero, and $\mathcal{D} = \varnothing$ is empty. Here, $r(s)$ represents the reward of reaching $s$. For any Markovian policy, the maximal return is $1/4$ (or $1/2^{d-1}$ for a depth-$d$ structure). In contrast, the optimal uncertainty-adaptive policy has a return of 1. This is achieved by maintaining a hypothesis set $\mathcal{C} = \{\mathcal{M}_1, \mathcal{M}_2, \mathcal{M}_3, \mathcal{M}_4\}$. After visiting a leaf state and observing reward $r \in \{0, 1\}$, update $\mathcal{C} \leftarrow \{\mathcal{M} \in \mathcal{C} : r_{\mathcal{M}}(s) = r\}$. Then each $r = 0$ removes one hypothesis from $\mathcal{C}$, and $r = 1$ identifies the true MDP. After at most four leaf visits, we have found the rewarding leaf and the return is 1. We can increase $d$ to make the performance gap (1 versus $1/2^{d-1}$) arbitrarily large. $\square$

> **Comparison Between Conventional RL and Bayesian RL**
>
> Conventional RL does not admit strictly uncertainty-adaptive policies, which do not give rise to reflective exploration. Instead, exploration is trial-and-error during training and only serves to fit a fixed policy. In contrast, Bayesian RL yields uncertainty-adaptive policies whose actions depend on the evolving belief over MDPs, thereby incentivizing information gathering and inducing reflective behaviors through belief updates at deployment time.

**Connection with Meta-RL.** Although our Bayesian RL framework is orthogonal to meta-RL, we discuss the connections between them here. The goal of meta-RL is to learn a fast adaptation procedure across tasks, e.g., by optimizing deployment performance after only a few iterations of updates. From this perspective, uncertainty-adaptive policy in Definition 3.1 can be viewed as performing in-context learning (instead of parameter updates), and our framework can be interpreted as learning to do in-context learning. However, our analysis focuses more on properties of policies that optimize the Bayesian RL objective, which enforce the Bayes-optimal solution to the exploration-exploitation trade-off. In other words, our framework explicitly maintains a belief over MDP and uses a belief-driven exploration strategy that rewards under the posterior.

## 5 METHOD

In the previous section, we analyzed the benefits of Bayesian RL and the role of self-reflection within the Bayesian RL framework. In what follows, we present a practical way to optimize the Bayesian RL objective in (3.1). The policy gradient for (3.1) is as follows, which differs from the conventional policy gradient by replacing the value under $\mathcal{M}^*$ with a posterior-weighted value:

$$\nabla_\theta \mathcal{J} = \mathbb{E}_{s_0, \pi_\theta}\left[\sum_{t=0}^{T-1} \nabla_\theta \log \pi_\theta(a_t \mid h_t) \cdot \mathbb{E}_{\mathcal{M}\sim p(\mathcal{M}|\mathcal{D}, h_t)}\left[Q_\mathcal{M}^{\pi_\theta}(h_t, a_t)\right]\right]. \tag{5.1}$$

Here, we use the state-action value $Q$ instead of calculating the advantage since the latter requires branched Monte Carlo rollouts at each step, which is computationally costly. By applying the Bayes rule, the posterior satisfies:

$$p(\mathcal{M} \mid \mathcal{D}, h_t) = p(\mathcal{M} \mid \mathcal{D}, s_{0:t}, a_{0:t-1}, r_{0:t-1}) \propto p(\mathcal{M} \mid \mathcal{D}, s_{0:t}) \cdot p(r_{0:t-1} \mid s_{0:t}, a_{0:t-1}, \mathcal{M}), \tag{5.2}$$

where $p(\mathcal{M}|\mathcal{D}, s_{0:t})$ conditions only on the CoT, excluding rewards, and can be modeled as the probability that the policy $\pi_\theta$, after training on $\mathcal{D}$, generates the solution $y_{s_0}^\mathcal{M}$. Interestingly, the second term $p(r_{0:t-1}|s_{0:t}, a_{0:t-1}, \mathcal{M})$ measures the likelihood of observing the rewards $r_{0:t-1}$ under $\mathcal{M}$ given the trajectory $s_{0:t}$. We follow Ghosh et al. (2022) to write $p(r_t|s_t, a_t, \mathcal{M}) \propto \exp(-\beta|r_t - r_\mathcal{M}(s_t, a_t)|)$ with a hyperparameter $\beta$ to obtain

$$p(r_{0:t-1} \mid s_{0:t}, a_{0:t-1}, \mathcal{M}) \propto \prod_{t'=0}^{t-1} p(r_{t'} \mid s_{t'}, a_{t'}, \mathcal{M}) = \prod_{t'=0}^{t-1} \exp\left(-\beta|r_{t'} - r_\mathcal{M}(s_{t'}, a_{t'})|\right), \tag{5.3}$$

where the proportionality holds since $p(r_{t'}|r_{0:t'-1}, s_{0:t}, a_{0:t-1}, \mathcal{M}) = p(r_{t'}|s_{t'}, a_{t'}, \mathcal{M})$.

Besides, the posterior-weighted value in (5.1) satisfies

$$\mathbb{E}_{\mathcal{M}\sim p(\mathcal{M}|h_t)}\left[Q_\mathcal{M}^{\pi_\theta}(h_t, a_t)\right] = \mathbb{E}_{\mathcal{M}\sim q(\mathcal{M}|s_0)}\left[Q_\mathcal{M}^{\pi_\theta}(h_t, a_t)\frac{p(\mathcal{M} \mid h_t)}{q(\mathcal{M} \mid s_0)}\right] = \sum_{i=0}^{|\mathcal{M}|} Q_{\mathcal{M}_i}^{\pi_\theta}(h_t, a_t)p(\mathcal{M}_i \mid h_t),$$

where the proposal $q(\mathcal{M}|s_0)$ is the uniform distribution over the support of plausible MDPs, defined w.r.t. the ground-truth answer and candidate answers extracted from the model's CoTs. We draw $|\mathcal{M}|$ CoT rollouts of $\pi_\theta$ on prompt $s_0$ to form $\{\mathcal{M}_i\}_{i=1}^{|\mathcal{M}|}$. The importance ratios are then self-normalized over $\{\mathcal{M}_i\}_{i=1}^{|\mathcal{M}|}$ and the ground-truth MDP $\mathcal{M}_0$ defined w.r.t. $y_{s_0}^*$. Substituting (5.2) and (5.3) into the above equation and letting $p(\mathcal{M}_i|s_{0:t})$ be the model's state-conditional belief gives

$$\mathbb{E}_\mathcal{M}\left[Q_\mathcal{M}^{\pi_\theta}(h_t, a_t)\right] = \sum_{i=0}^{|\mathcal{M}|} \underbrace{Q_{\mathcal{M}_i}^{\pi_\theta}(h_t, a_t)}_{\text{value in } \mathcal{M}_i} \underbrace{\pi_\theta(y_{s_0}^{\mathcal{M}_i}|s_t + \text{</think>})}_{\text{model plausibility for } \mathcal{M}_i} \prod_{t'=0}^{t-1} \underbrace{\exp\left(-\beta|r_{t'} - r_{\mathcal{M}_i}(s_{t'}, a_{t'})|\right)}_{\text{consistency b/w observed \& } \mathcal{M}_i\text{'s reward}}, \tag{5.4}$$

where $r_{t'}$ is the actual observed reward as defined in (3.2), and $r_{\mathcal{M}_i}, Q_{\mathcal{M}_i}^{\pi_\theta}$ are defined in (3.3) w.r.t. the hypothesis MDP $\mathcal{M}_i$. Since rewards are not available at deployment time, they are not explicitly provided as inputs to the policy. Instead, maximizing the Bayesian RL objective forces the parameters $\theta$ to encode a prediction of the reward as a function of state, thus absorbing the dependence on rewards into $\theta$. For clarity, we have omitted the normalization constant when computing $p(\mathcal{M}_i|h_t)$ from the last two terms. Algorithm 1 provides the unbatched version of the pseudocode.

---

**Algorithm 1** Bayes-Adaptive RL for LLM Reasoning (BARL)

1: **Input:** LLM $\pi_\theta$, prompt set $\{s_0\}$ with ground-truth answers $\{y^*_{s_0}\}$.
2: **for** each prompt $s_0$ in the training data **do**
3:      Sample $|\mathcal{M}|$ CoTs from $\pi_\theta$ on $s_0$.
4:      Extract the $|\mathcal{M}|$ candidate answers from each CoT to form a candidate set $\{y^{\mathcal{M}_i}_{s_0}\}_{i=1}^{|\mathcal{M}|}$.
5:      **for** each of the $|\mathcal{M}|$ CoTs **do**
6:          Calculate posterior-weighted value with (5.4) for each timestep $t$ and update $\theta$ with (5.1).

---

BARL offers a principled framework for stitching together plausible strategies, analogous to linearizing best-of-N reasoning, but with guidance on *when* and *how* LLMs should reflectively explore.

**Remark 5.1.** BARL maximizes the weighted sum of values defined over each hypothesis MDP $\mathcal{M}_i$. The first weighting term $\pi_\theta(y^{\mathcal{M}_i}_{s_0}|\cdot)$ captures LLM's state-conditional belief in the plausibility of $\mathcal{M}_i$. The second product weighting term accumulates the discrepancy between predicted rewards $r_{\mathcal{M}_i}(s_{t'}, a_{t'})$ and observed rewards $r_{t'}$, which serves as a reflective signal for strategy switching by downweighting hypotheses that have high belief probabilities but are unlikely to be optimal.

## 6 How BARL Helps Generalization: A Didactic Example

Now we present a didactic example to show how BARL facilitates eest-time generalization. Consider the action space $\mathcal{A}$ that consists of three tokens, $\{0, 1, 2\}$, with one token generated at each timestep. The objective is to repeat the prompt token three times consecutively within 29 timesteps ($3^3 + 2$ is the minimal length to include all unique triplets). The prompt token is 0 or 1 during training, which does not have coverage for 2. After training, the policy is deployed, where we consider the prompt token of 2 as the evaluation task. Episodes terminate when receiving a 1 reward. This setup is illustrated in Figure 2.

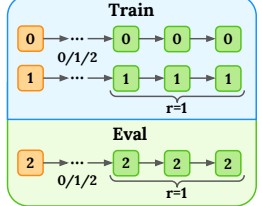

Figure 2: Setup: repeating the prompt token (orange) three times receives a 1 reward.

This setup mirrors LLM reasoning, where the goal is to not only learn specific strategies (here, generating particular triplets), but also general problem-solving abilities, such as when and how to switch to new strategies. These capabilities are essential for handling distribution shifts between training and eval, a common challenge when developing LLMs.

We use a 2-head transformer encoder followed by a linear layer as the policy and train it using the policy gradient from conventional RL and from BARL (5.1). For RL, the state-action value $Q^{\pi_\theta}$ in the policy gradient is 1 only when $s_{0:T-1}$ contains the rewarding triplet $\arg\max_{\text{tri}} r(\text{tri})$ of the ground-truth MDP, such as 000 or 111 during training. For BARL, we set $\beta = \infty$ so that $\prod_{t'=0}^{t-1} \exp(-\beta|r_{t'} - r_{\mathcal{M}_i}(s_{t'}, a_{t'})|) = \mathbb{1}(\arg\max_{\text{tri}} r_{\mathcal{M}_i}(\text{tri}) \notin s_{0:t})$, i.e., the product is 0 when the rewarding triplet of $\mathcal{M}_i$ already appears in $s_{0:t}$ (since $r_{t'} = 0$ for unterminated sequences) and 1 otherwise. For the policy's state-conditional belief $p(\mathcal{M}|s_{0:t})$, sampling from it is equivalent to sampling $a_t \sim \pi_\theta(\cdot|s_t)$, with the associated $\mathcal{M}$ satisfying $\arg\max_{\text{tri}} r_{\mathcal{M}}(\text{tri}) = s_{t-1:t+1}$. Therefore,

$$\mathbb{E}_{\mathcal{M}}\big[Q^{\pi_\theta}_{\mathcal{M}}(s_t, a_t)\big] = \mathbb{E}_{\mathcal{M} \sim p(\mathcal{M}|s_{0:t})}\big[Q^{\pi_\theta}_{\mathcal{M}}(s_t, a_t)\mathbb{1}(\arg\max_{\text{tri}} r_{\mathcal{M}}(\text{tri}) \notin s_{0:t})\big] = \mathbb{E}_\pi\big[\mathbb{1}(s_{t-1:t+1} \notin s_{0:t})\big].$$

The above formulation incentivizes the policy to eliminate hypotheses and switch to new strategies (i.e., new triplets) when the current strategy has been invalidated by earlier attempts up to step $t$, which is illustrated in Figure 3. This form of adaptive exploration provides a minimalist instantiation of BARL in synthetic settings. The difference between the above equation and (5.4) arises because the agent here is aware of the zero reward associated with unfinished episodes.

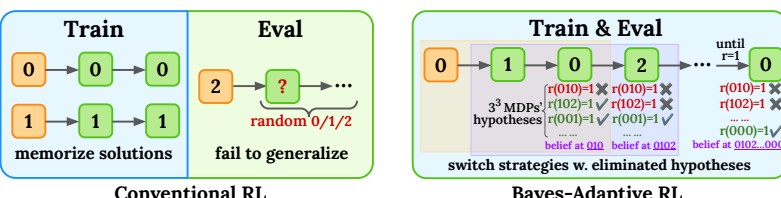

Figure 3: Comparison between conventional RL and BARL in this didactic example.

We report the results in Figure 4, where accuracies are averaged over 50 completions and the shadow regions are the standard deviation across 3 independent model training runs. The results show that RL quickly finds and memorizes the training solutions but fails to generalize when deployed to the evaluation tasks. In contrast, Bayes-Adaptive RL significantly increases the evaluation accuracies. Furthermore, the accuracy and convergence rate improve when given prior knowledge that rewarding triplets are repeated patterns, i.e., $|\mathcal{M}| = 3$ with $r_{\mathcal{M}_1}(000) = r_{\mathcal{M}_2}(111) = r_{\mathcal{M}_3}(222) = 1$. This highlights the advantage of more informative candidate sets, underscoring the importance of balancing *diversity* and *plausibility* of the candidates. Specifically, they should be diverse enough to capture deployment-time uncertainty, yet constrained to only the most plausible candidates to shrink the hypothesis space.

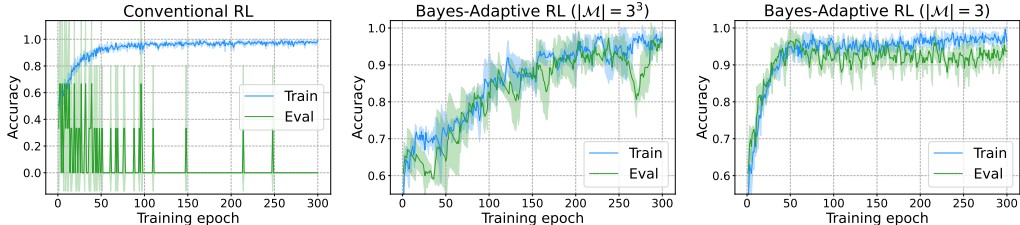

Figure 4: Conventional RL (REINFORCE) memories training solutions and poorly generalizes beyond the training prompts. BARL performs well when deployed to the evaluation tasks, and improves with more informative candidate sets.

## 7 EXPERIMENTS

### 7.1 EXPERIMENT SETUPS

In addition to the synthetic experiment, we evaluate BARL on math tasks. We implement BARL across various models, including Qwen2.5-Math-1.5B, Qwen2.5-Math-7B (Yang et al., 2024), and DeepSeek-R1-Distill-Llama-8B (Guo et al., 2025). Training is conducted on the Big-Math dataset (Albalak et al., 2025), and evaluation is performed on four benchmarks: GSM8K (Cobbe et al., 2021), MATH (Hendrycks et al., 2021), CollegeMath (Tang et al., 2024), and OlympiadBench (He et al., 2024). During training, the maximum prompt length is set to 512 and the maximum response length is set to 1024. For BARL, we set $\beta = 1$ and $|\mathcal{M}| = 5$ in Algorithm 1.

We compare BARL against two RL baselines that span outcome- and process-reward RL. For the outcome-reward GRPO baseline, we set its group size to 5 for a fair comparison with BARL and, after performing a grid search over the KL-divergence coefficients $[0, 0.001, 0.005, 0.01]$, adopt 0.005 as it yields the best overall performance across all benchmarks. For the process-reward baseline, we adapt a variant of MRT (Qu et al., 2025) by integrating the progress reward defined in (3.2) into the outcome reward, which we refer to as *progress* in the following sections. For all algorithms, we set the training and rollout batch sizes to 128 and 1024, respectively. We train the Qwen and Llama models for 110 and 60 iterations, respectively, defined w.r.t. the rollout batches. The temperature during online sampling is 1.0 and is 0.0 during evaluation. For both BARL and the progress baseline, we set the number of tokens for each reasoning step as 128.

### 7.2 EXPERIMENT RESULTS

We report the pass@1 accuracies in Table 1. All models are trained using three random seeds, and we calculate the mean and standard error of the resulting accuracies. For each algorithm, we report the results of the checkpoint that has the overall best performance on all benchmarks.

We observe that BARL achieves higher accuracies across most benchmarks and models. It consistently outperforms the two conventional RL baselines in terms of average accuracy, with the most significant gains observed on challenging benchmarks that demand effective exploration. Notably, BARL delivers these gains with minimal computational overhead, which comes from calculating the probabilities of candidate answers at the end of each step, reusing the prefix cache from CoT generations. We defer the curves of accuracies and response lengths to Appendix B.1 and B.2.

| Model | GSM8K | MATH | College | Olympiad | AIME | AMC | Average |
|---|---|---|---|---|---|---|---|
| Qwen-1.5B | 40.0 | 34.1 | 6.6 | 21.8 | 16.7 | 32.5 | 25.3 |
| GRPO | 83.9(±0.5) | 71.5(±0.4) | 45.1(±0.3) | 31.6(±0.4) | **17.8**(±0.9) | 55.8(±0.7) | 51.0(±0.3) |
| Progress | 84.8(±0.6) | 72.1(±0.4) | 45.9(±0.3) | 35.5(±0.3) | 14.4(±0.9) | 55.8(±0.7) | 51.4(±0.2) |
| BARL | **85.8**(±0.1) | **72.7**(±0.3) | **46.8**(±0.2) | **35.8**(±0.3) | **17.8**(±0.9) | **60.8**(±0.7) | **53.3**(±0.4) |
| Qwen-7B | 59.1 | 53.7 | 21.9 | 19.0 | 13.3 | 42.5 | 34.9 |
| GRPO | 90.3(±0.1) | 77.6(±0.4) | **47.0**(±0.3) | 38.5(±0.2) | 24.5(±0.4) | 65.0(±1.2) | 57.1(±0.2) |
| Progress | 91.1(±0.3) | 78.8(±0.1) | 46.7(±0.1) | 41.1(±0.2) | 24.5(±0.4) | 65.4(±1.4) | 57.9(±0.2) |
| BARL | **91.7**(±0.2) | **79.2**(±0.3) | 47.5(±0.2) | **42.0**(±0.4) | **29.0**(±0.4) | **66.7**(±1.4) | **59.4**(±0.3) |
| Llama-8B | 82.0 | 65.7 | 35.8 | 26.5 | 10.0 | 42.5 | 43.7 |
| GRPO | 85.7(±0.5) | **74.3**(±0.4) | 39.6(±0.3) | 36.0(±0.5) | 16.7(±0.8) | 60.4(±0.3) | 52.1(±0.4) |
| Progress | **85.9**(±0.4) | 73.9(±0.4) | 39.8(±0.5) | 35.3(±0.2) | 15.0(±0.8) | 55.8(±0.7) | 51.0(±0.4) |
| BARL | 85.4(±0.6) | 73.9(±0.6) | **40.4**(±0.3) | **37.2**(±0.5) | **17.8**(±0.9) | **61.7**(±0.7) | **52.7**(±0.4) |

Table 1: Mean and standard error of the accuracies over three independent training runs.

## 7.3 ABLATION STUDIES

**Token Efficiency.** We evaluate the token efficiency of BARL and baseline methods by measuring the total numbers of tokens required to solve a problem. Specifically, in Figure 5, we plot the pass@k accuracies and the corresponding average numbers of tokens as a proxy for per-token performance. In all the following ablations, we mainly analyze models fine-tuned from Qwen2.5-Math-1.5B. We vary k from 1 to 6 and set the temperature to 1.0, but observe that the GRPO and base models are less robust under sampling, resulting in decreased pass@1 performance (see Appendix B.3). To account for this, we combine greedy decoding outputs with sampled outputs when computing token usage and accuracy. We omit the base model from the plots due to its significantly higher token consumption and lower asymptotic accuracy. We find that BARL achieves higher accuracies with substantially fewer tokens, requiring up to **1.63x** fewer average tokens than the progress baseline, **2x** fewer than GRPO, and over **10x** fewer than the base model.

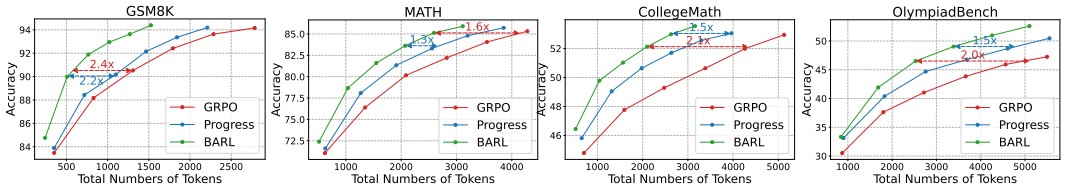

Figure 5: BARL achieves higher pass@k accuracies with fewer total numbers of tokens.

**Frequency of Self-Reflection Behaviors.** To qualitatively assess the improved token efficiency of BARL, we analyze the frequency of reflective behaviors across problems of varying difficulty levels, as shown in Figure 7.3, with GSM8K and MATH in dashed and solid lines, respectively. For each problem, we sample 6 responses per model and define its difficulty level by the number of incorrect responses. We use keyword-based detections (Liu et al., 2025; Yeo et al., 2025) to identify if self-reflections appear in a response. A problem is considered to exhibit self-reflection if at least one of its responses is identified. It can be observed that both the base and the BARL models display fewer reflections on easier problems. The base model exhibits an overall higher

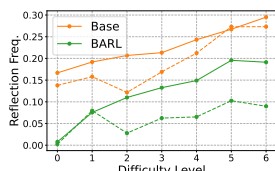

Figure 6: Reflection freq. on GSM8K (dashed) and MATH (solid) problems.

frequency of reflections despite achieving lower accuracies. This result reveals the weak correlation between the performance of LLMs and the response length or the frequency of reflections. Rather, the effectiveness of thinking tokens and the efficiency of explorations are the determining factors, which we will study in the following. Notably, our conclusions are not weakened by the potential inaccuracies of keyword-based detections. The fact that the base model triggers more "reflection" tokens yet underperforms BARL suggests that it has acquired superficial stylistic reflection patterns during pre-training (e.g., from long-CoT annealing data) without effective exploration, whereas BARL yields more effective reflective exploration rather than more frequent but less effective stylistic reflections.

**Effectiveness of CoTs.**    We measure the effectiveness of the CoTs produced by different models by calculating the average Bayesian state-action values at all timesteps, which naturally captures both the exploration and the exploitation aspects of the actions. Unlike standard Q-values, the Bayesian Q-value not only incorporates the expected returns (exploitation) but also captures the value of information gained through belief updates (exploration).

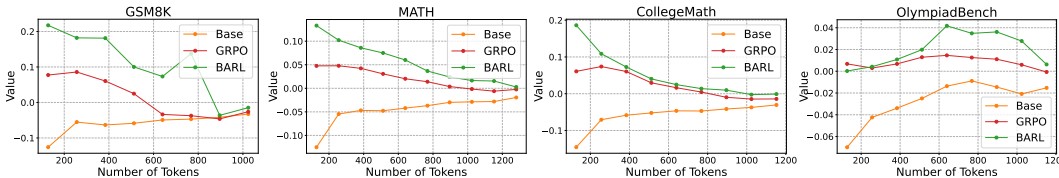

Figure 7: Ablation on how effective the CoTs explore and exploit, measured by the Bayesian values.

The results are reported in Figure 7. We observe that the actions from the BARL model exhibit consistently higher Bayesian values compared to those of the GRPO and base models, indicating more effective exploration and exploitation. On more challenging benchmarks such as OlympiadBench, exploratory gains peak midway through the CoTs after an early phase of uncertainty reduction. Moreover, the result also explains our earlier observations on token efficiency and reflective behaviors. Although the base model exhibits more self-reflections, these are likely superficial or stylistic patterns due to their low exploration efficiency for gathering informative contexts during evaluation.

**Conventional RL Optimality.**    We train a length-controlled (LC) GRPO with a maximum 32 response length over multiple epochs. Figure 8 shows the evolution of the training accuracy and response length. The rapidly decreasing length of LC GRPO indicates that it learns to skip CoT generation and emit only the final answer. Its asymp-

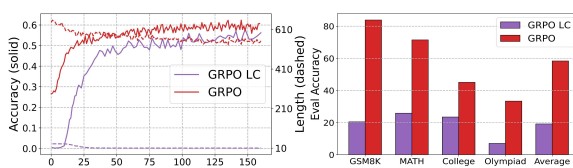

Figure 8: Training accuracies, lengths and eval results.

totic training accuracy matches that of GRPO (max length 1024). This result supports Theorem 4.1: conventional RL can achieve optimality by merely memorizing solutions without reflective reasoning. Such policies, however, generalize poorly during evaluation.

> **Key Experiment Findings**
>
> BARL consistently outperforms conventional RL baselines with superior token efficiency. Performance correlates with the effectiveness of reflective explorations, rather than their frequency. Optimality in conventional RL can be attained by policies that memorize training solutions yet fail to generalize, with no guarantees on the emergence of self-reflections.

# 8    CONCLUSION

LLMs trained via RL have exhibited emergent behaviors such as self-reflections, which offers a potential way for test-time scaling by backtracking to a previous state and generating a different action distribution with updated context. However, conventional RL training neither ensures the emergence of reflective exploration nor explains its benefits. The optimal RL policies are typically Markovian and have no incentive to enrich a revisited state with additional context since they depend on the history only through the state. Instead, RL exploration is confined to the training phase to identify action sequences that maximize cumulative rewards, and resorts to pure exploitation when the policy parameters are frozen after deployment. We propose to ground reflective exploration with a Bayesian RL objective that maximizes expected deployment-time return under a posterior distribution over MDPs induced by the training data. To maximize this objective, we propose BARL, which provides step-level guidance for when and how to engage in reflective exploration. BARL enables efficient exploration through hypothesis elimination and strategy switching. Our experiments are conducted on both synthetic and mathematical reasoning tasks, where we show that BARL outperforms conventional RL algorithms for evaluation benchmarks, and its exploration is more efficient.

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

# A PROOFS

## A.1 PROOF OF THEOREM 4.1

*Proof.* Let $\pi^*$ be any optimal policy (possibly history-dependent). Suppose $\pi^*$ prescribes different action distributions at the same state $s$ depending on the history that led to $s$.

Let $\delta_1$ and $\delta_2$ be two such distributions with $\mathbb{E}_{a \sim \delta_1} Q^*(s, a) > \mathbb{E}_{a \sim \delta_2} Q^*(s, a)$, where $Q^*$ is the optimal value in RL. Then construct $\pi'$ that behaves like $\pi^*$ everywhere except it always uses $\delta_1$ whenever the state is $s$ (ignoring history). The future trajectories from any visit to $s$ are stochastically identical under $\pi^*$ and $\pi'$, except that $\pi'$ gets strictly higher expected return from every time $s$ is visited after histories that would have triggered $\delta_2$ under $\pi^*$.

Thus, $V^{\pi'}(s_0) > V^{\pi^*}(s_0)$ for some state, contradicting optimality of $\pi^*$. Therefore, this optimal policy must prescribe the same action distribution at state $s$ regardless of history. In other words, there exists an optimal Markovian policy. In the following, we will prove by contradiction that policies that exhibit reflective exploration cannot be Markovian.

Assume $\pi$ is a Markovian policy. Then there exists a mapping $\mu : \mathcal{S} \to \Delta(\mathcal{A})$ such that, for every history $h_t$, $\pi(\cdot \mid h_t) = \mu(\cdot \mid s(h_t))$. Define the process $X_t := s(h_t)$ and set $\upsilon := \mu$. By construction, $\pi(\cdot \mid h_t) = \upsilon(\cdot \mid X_t)$ for all $t$.

According to Definition 3.2, there is a measurable $\psi$ such that $X_t = \psi(\varphi(s_t))$ and thus $s(h_t) = \psi(\varphi(s(h_t)))$. Let $s, s'$ be reachable states and $\varphi(s) = \varphi(s')$. Then $s = \psi(\varphi(s)) = \psi(\varphi(s')) = s'$.

Therefore, there exist $t_1 > t_2$ and histories $h_{t_1}, h_{t_2}$ such that

$$\varphi(s(h_{t_1})) = \varphi(s(h_{t_2})), \qquad \pi(\cdot \mid h_{t_1}) \neq \pi(\cdot \mid h_{t_2}).$$

Thus, we obtain $s(h_{t_1}) = s(h_{t_2}) =: s_*$. But then, using the Markov property of $\pi$, we have $\pi(\cdot \mid h_{t_1}) = \mu(\cdot \mid s_*) = \pi(\cdot \mid h_{t_2})$, which contradicts $\pi(\cdot \mid h_{t_1}) \neq \pi(\cdot \mid h_{t_2})$. Therefore, $\pi$ is not Markovian. $\square$

## A.2 PROOF OF THEOREM 4.2

Before the proof, we first provide several useful definitions and lemmas.

**Definition A.1** (Epistemic POMDP). We define the *epistemic POMDP* $\mathcal{P} := (\mathcal{Z}, \mathcal{O}, \mathcal{A}, P, r, \rho)$ where the hidden state space as $\mathcal{Z} := \mathcal{S} \times \mathcal{M}$, with state $z_t = (s_t, M)$; action space $\mathcal{A}$; observation space $\mathcal{O} := \mathcal{S}$ with an observation map $O((s, M)) = s$; transition kernel $P\big((s_{t+1}, M') \mid (s_t, M), a_t\big) = \mathbf{1}\{M' = M\} P_M(s_{t+1} \mid s_t, a_t)$; reward $r((s, M), a) = r_M(s, a)$; and initial state distribution $\rho((s_0, M)) := p(M \mid D) \rho_M(s_0)$, where $\rho_M$ is the initial-state distribution in $M$. For a policy $\pi$, let $J_\mathcal{P}(\pi)$ denote its expected return in this POMDP.

**Lemma A.2** (Equivalence of Objectives). For any policy $\pi$, $\pi$ is Bayes-optimal if and only if it is optimal for the epistemic POMDP $\mathcal{P}$, i.e.,

$$J_\mathcal{P}(\pi) = J_{\text{Bayes}}(\pi).$$

*Proof.* By construction of the initial distribution,

$$(s_0, M) \sim \rho \iff M \sim p(M \mid D), \ s_0 \sim \rho_M.$$

Conditioned on $M$, the state transitions and rewards in $\mathcal{P}$ coincide exactly with those of the MDP $M$ under policy $\pi$, and the observation $o_t = s_t$ carries no additional information beyond $s_t$. Thus

$$J_\mathcal{P}(\pi) = \mathbb{E}_{M \sim p(M|D)}[J_M(\pi)] = J_{\text{Bayes}}(\pi),$$

which proves the claim. Similar results are also presented in Singh et al. (1994); Duff (2002). $\square$

Now we are ready to prove Theorem 4.2.

*Proof of Theorem 4.2.* By Lemma A.2, Bayes-optimal policies coincide with optimal policies for the epistemic POMDP $\mathcal{P}$.

It is a standard result in POMDP theory that there exists an optimal policy which is a function only of the belief state $\beta_t$, where

$$\beta_t(z) := \mathbb{P}(z_t = z \mid h_t, D).$$

Thus there exists a policy $\pi^\star$ and a measurable mapping $\widetilde{\mu}^\star$ such that

$$\pi^\star(a_t \mid h_t) = \widetilde{\mu}^\star(a_t \mid \beta_t).$$

In the epistemic POMDP, the hidden state is $z_t = (s_t, M)$ and the observation is $o_t = s_t$. Therefore the belief factors as

$$\beta_t(s, M) = \mathbf{1}\{s = s_t\}\, p(M \mid h_t, D),$$

so $\beta_t$ is uniquely determined by the pair $(s_t, p(\cdot | h_t, D))$. Since the offline posterior $p(M|D)$ is fixed, the mapping $b(h_t)(M)$ is in one-to-one correspondence with $p(M|h_t, D)$, and hence with $\beta_t$. Thus there exists a measurable function $\mu^\star$ such that for all $h_t$,

$$\widetilde{\mu}^*(a_t \mid \beta_t) = \mu^\star(a_t \mid s_t, b(h_t)).$$

Therefore, we obtain

$$\pi^\star(a_t \mid h_t) = \widetilde{\mu}^\star(a_t \mid \beta_t) = \mu^\star(a_t \mid s_t, b(h_t)),$$

so $\pi^*$ is uncertainty-adaptive. Since $\pi^*$ is optimal for $\mathcal{P}$, it is Bayes-optimal for $J_{\text{Bayes}}$ by Lemma A.2, completing the proof. $\qquad\square$

### A.3   PROOF OF THEOREM 4.3

*Proof.* Consider a full binary decision structure of depth $T^*$, with leaf set $\mathcal{L}$ of size $|\mathcal{L}| = 2^{T^*}$. The states are the tree nodes, with the initial state fixed as the root node. The actions include the moves from the parent nodes to the child nodes as well as a resetting action from the leaf node to the root node. The rewards are not known and differ in different MDP hypotheses. Specifically, the reward is defined as $r_{\mathcal{M}_i}(s) = \mathbb{1}(s = s_i)$, where $s_i$ is a unique leaf node for $\mathcal{M}_i$. The prior of MDPs is $p(\mathcal{M}_i) = 1/|\mathcal{L}|$, where $i = 1, \cdots, |\mathcal{L}|$. The cumulative return is undiscounted with the minimum traverse steps as the horizon $T$. The episode terminates once the agent receives a 1 reward.

For any Markovian policy $\pi$, define $f(s) = p(\exists t \geq 0 : s_t = s \mid \pi)$. We define $p_s$ as the probability that $\pi$ goes left at an internal node $s$. By the Markov property, for any left and right children $s_L$ and $s_R$ of node $s$, it holds that $f(s_L) = f(s)p_s$ and $f(s_R) = f(s)(1-p_s)$. Thus, $f(s_L)+f(s_R) = f(s)$. By a simple induction on depth, it follows that at every depth $d$, $\sum_{s:\text{depth}(s)=d} f(s) = 1$. In particular, $\sum_{l \in \mathcal{L}} f(l) = 1$. Since the total return under $\mathcal{M}_i$ is $V(\pi \mid \mathcal{M}_i) = p(\pi \text{ ends at leaf } l_i) = f(l_i)$, the expected return for the optimal Markovian policy is

$$\frac{1}{|\mathcal{L}|} \sum_{i=1}^{|\mathcal{L}|} V(\pi \mid \mathcal{M}_i) = \frac{1}{2^{T^*}} \sum_{l \in \mathcal{L}} f(l) = \frac{1}{2^{T^*}}.$$

We construct the uncertainty-adaptive policy as follows. Let $\mathcal{C} = \{\mathcal{M}_1, \ldots, \mathcal{M}_{|\mathcal{L}|}\}$ be the set of models that are still consistent with the observed rewards in $h_t$. When the agent visits a leaf $s$ and observes reward $r \in \{0, 1\}$, we update $\mathcal{C} \leftarrow \{\mathcal{M} \in \mathcal{C} : r_{\mathcal{M}}(s) = r\}$. For each $i$, there is exactly one leaf $s$ with $r_{\mathcal{M}_i}(s) = 1$. Hence, under any true MDP $\mathcal{M}^*$, after at most $|\mathcal{L}|$ visits to leaves we must have found the reward leaf, and obtain a return of 1. $\qquad\square$

## B   EXPERIMENT DETAILS

### B.1   EVALUATION RESULTS: ACCURACIES

In this section, we provide the evaluation accuracy results for Qwen2.5-Math-1.5B, Qwen2.5-Math-7B, and DeepSeek-R1-Distill-Llama-8B in Figure 9, 10, and 11, respectively. All models are trained using the same random seed and hyperparameters as described in Section 7.1. It can be observed that BARL outperforms the two RL baselines in terms of both accuracy and convergence rate on most of the reported benchmarks.

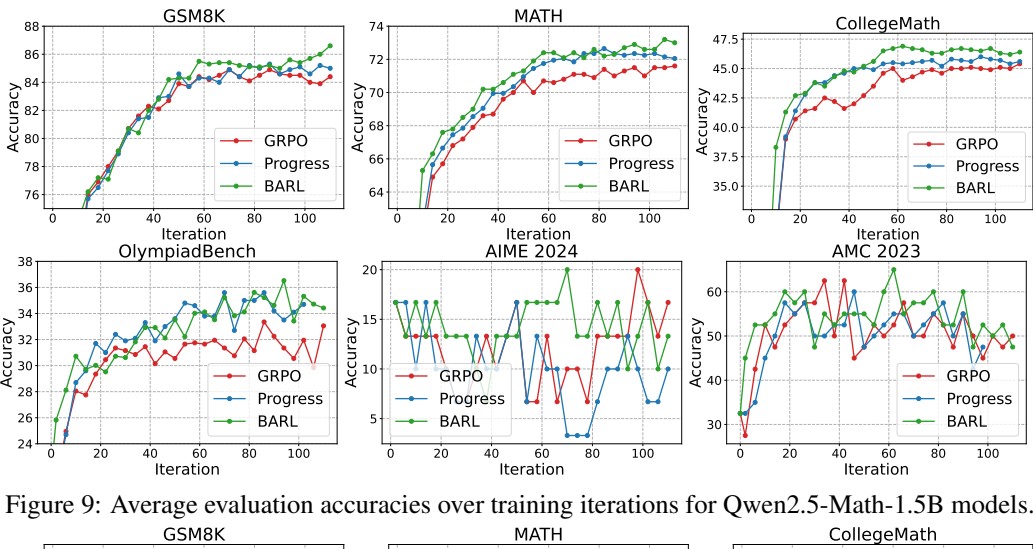

Figure 9: Average evaluation accuracies over training iterations for Qwen2.5-Math-1.5B models.

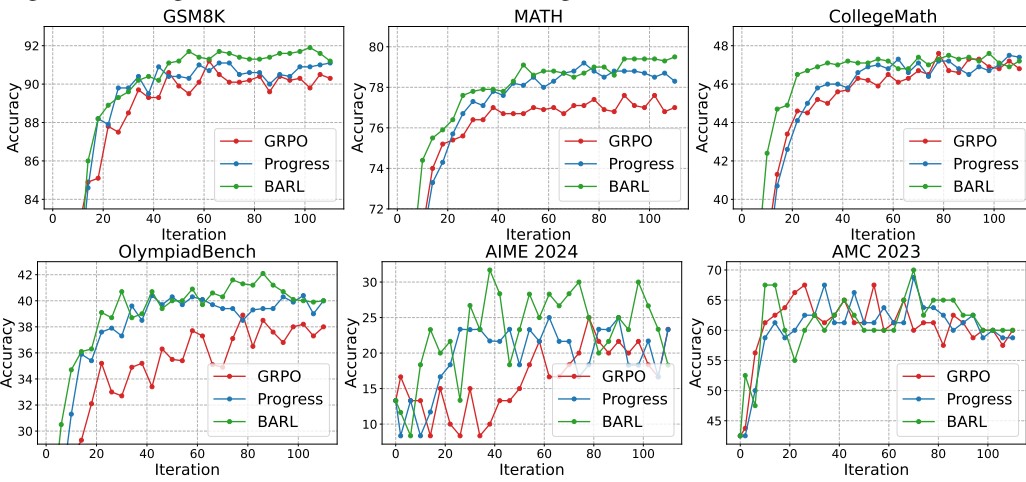

Figure 10: Average evaluation accuracies over training iterations for Qwen2.5-Math-7B models.

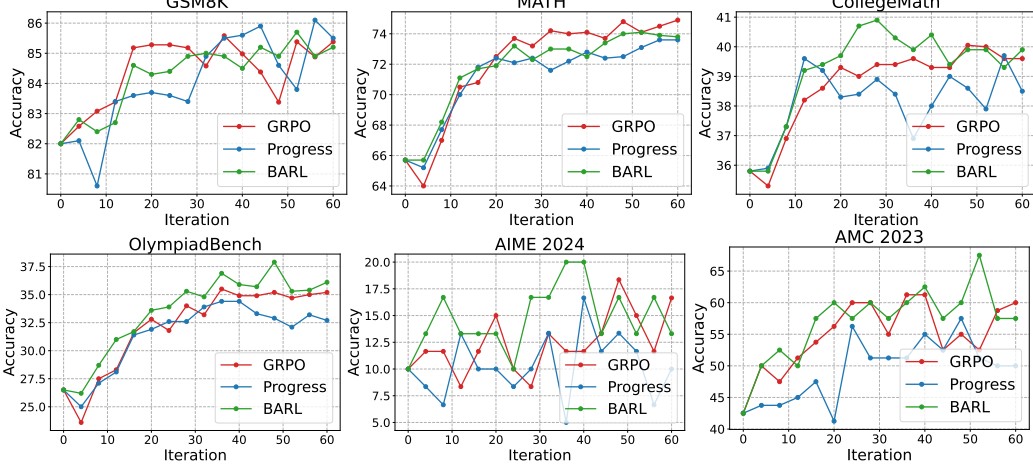

Figure 11: Average evaluation accuracies over training iterations for R1-Distill-Llama-8B models.

## B.2 EVALUATION RESULTS: RESPONSE LENGTHS

In this section, we present the evolution of evaluation response lengths over training iterations for Qwen2.5-Math-1.5B, Qwen2.5-Math-7B, and DeepSeek-R1-Distill-Llama-8B, shown in Figures 12, 13, and 14, respectively. Across most benchmarks, response lengths tend to decrease as training progresses for all algorithms. The response length during training has a very similar trend to that during evaluation. This trend arises because all three models exhibit reflective behaviors, such as self-evaluation and backtracking, that introduce redundant tokens and lengthen responses. As shown in Figure 7, these behaviors are likely superficial or stylistic patterns with limited effectiveness. An exception is AIME, where some models maintain consistently long responses due to the benchmark's intrinsic requirement for extended reasoning, even under optimal non-reflective policies.

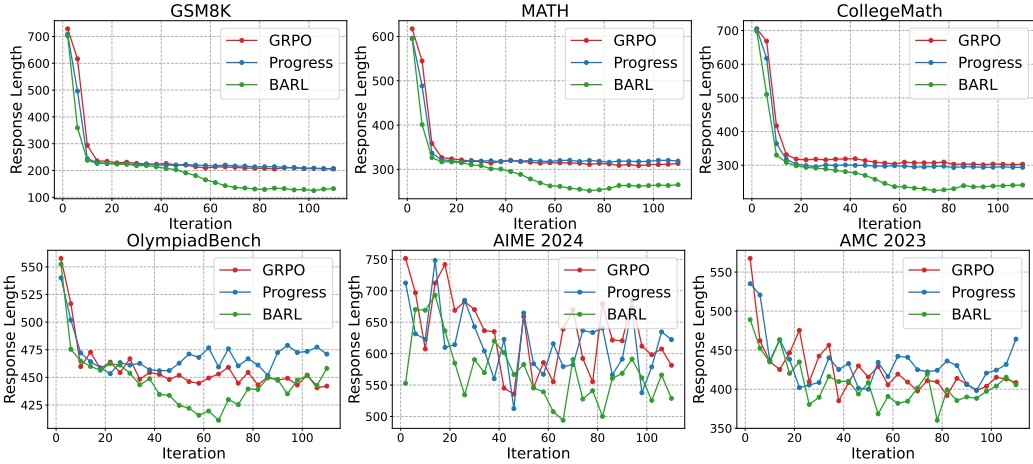

Figure 12: Average evaluation response lengths over training iterations for Qwen2.5-Math-1.5B models.

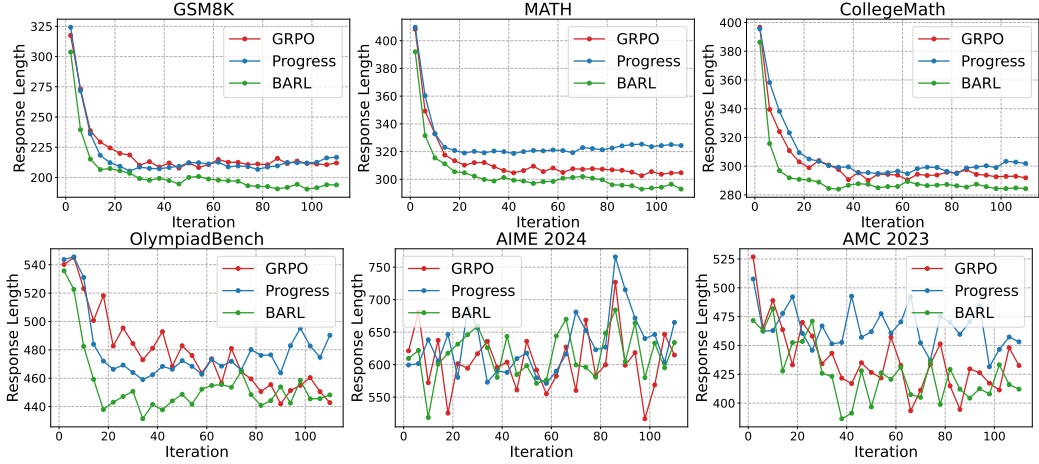

Figure 13: Average evaluation response lengths over training iterations for Qwen2.5-Math-7B models.

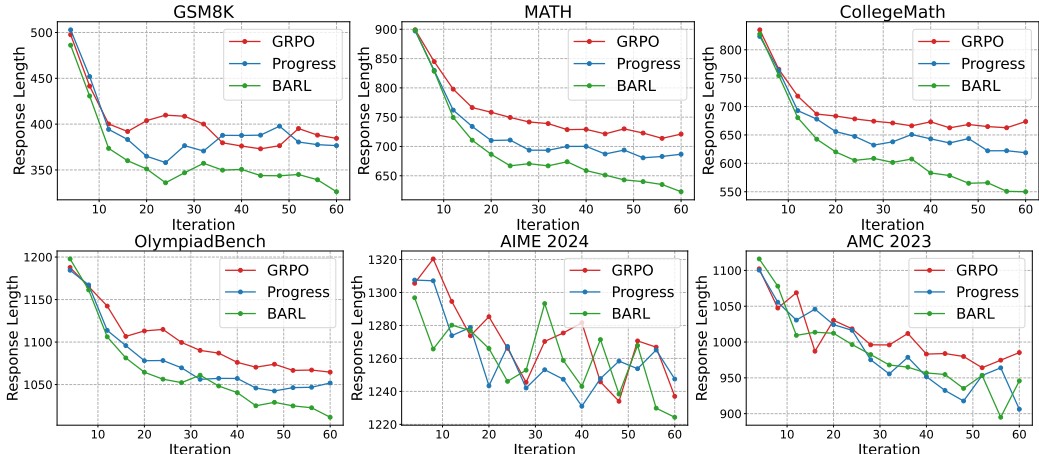

Figure 14: Average evaluation response lengths over training iterations for R1-Distill-Llama-8B models.

Since the models used in the main experiments already exhibit lengthy CoTs with reflective patterns, we further implement BARL on the Llama-3.2-3B-Instruct model, which displays fewer self-reflections. The results are presented in Figure 15. Initially, the response length decreases as the base model tends to produce excessively long reasoning traces, often exceeding ten steps, which are pruned during early training. Subsequently, the response length increases, as a result of plausible strategy stitching.

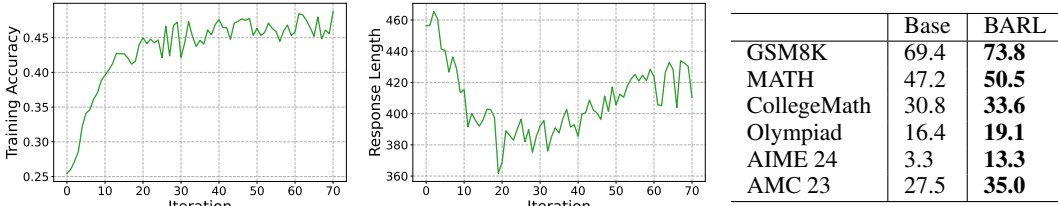

Figure 15: Results of BARL fine-tuned on Llama-3.2-3B-Instruct. **(Left)** Training accuracy and **(Middle)** response length. **(Right)** Evaluation results.

### B.3 TOKEN EFFICIENCY WITHOUT GREEDY DECODING OUTPUTS

In Figure 16, we present the pass@k accuracies from an ablation similar to Section 7.3, except that greedy decoding outputs are excluded when computing token counts and accuracies. We observe that the base and GRPO models are less robust under a sampling temperature of 1.0, resulting in significantly lower pass@1 accuracies compared to greedy decoding. This degradation may stem from the fragility of their CoTs, which often exhibit stylistic but unproductive self-reflection and backtracking behaviors.

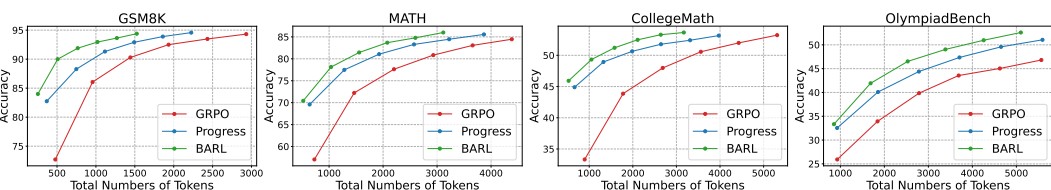

Figure 16: Ablation on token efficiency and pass@k accuracies with sampling temperature= 1. GRPO and the base models are less robust to temperatures.

### B.4 SOME UNSUCCESSFUL ATTEMPTS

As a straightforward implementation of the Bayes-Adaptive RL policy gradient in (5.1), we explored using value ensembles to estimate the posterior-weighted value $\mathbb{E}_{\mathcal{M} \sim p(\mathcal{M}|h_t)}[Q_{\mathcal{M}}^{\pi_\theta}(s_t, a_t)]$. Specifically, we trained an ensemble of state-action value functions to capture epistemic uncertainty. We experimented with two approaches to constructing the ensemble: (1) fine-tuning multiple linear value heads on disjoint data subsets, each paired with chain-of-thought (CoT) trajectories and outcome rewards; and (2) applying Bayesian LoRA. However, both methods failed to effectively capture epistemic uncertainty, likely because they fine-tune only a small subset of LLM parameters, which is insufficient to fully represent the uncertainty. While maintaining independent value models may better capture this uncertainty, doing so incurs substantial computational cost. We leave the development of more efficient implementations to future work.

## C STATEMENT ON THE USE OF LARGE LANGUAGE MODELS

We use LLMs only to polish the paper, such as improving clarity and grammar, without altering its substance or original ideas.

