# OpenReview forum: "Beyond Markovian: Reflective Exploration via Bayes-Adaptive RL for LLM Reasoning"
_ICLR.cc/2026/Conference — ICLR 2026 Poster_

### Official Review · Reviewer_QdQn · 2025-10-27

**Soundness:** 2
**Presentation:** 3
**Contribution:** 3
**Rating:** 6
**Confidence:** 3

**Summary:**

This paper addresses the limitations of standard Markovian RL in explaining or guaranteeing the emergence of reflective reasoning (like backtracking and error correction) observed in Large Language Models (LLMs) trained for complex tasks. It proposes recasting the problem within the Bayes-Adaptive RL framework, which explicitly optimizes for test-time performance under uncertainty about the underlying MDP. This Bayesian approach naturally incentivizes exploration to gather information and reduce uncertainty, leading to adaptive policies that update beliefs based on outcomes. The authors introduce BARL, an algorithm where the LLM maintains a posterior over MDP hypotheses, weighting actions by belief and penalizing discrepancies between predicted and observed rewards to guide strategy switching and reflective exploration. Empirical results on synthetic and math reasoning tasks show BARL outperforms Markovian RL baselines in accuracy and token efficiency, demonstrating the benefits of principled, belief-driven exploration over fixed, deterministic policies.

**Strengths:**

- The paper proposes a compelling Bayes-Adaptive RL framework that extends standard Markovian RL. This Bayesian approach provides a principled way to encourage exploration and generalization for LLM reasoning, moving beyond policies that might simply memorize training solutions.
- The core concepts are well-supported through both theoretical justification and empirical validation. The inclusion of a didactic synthetic example significantly aids in illustrating the core mechanism and benefits of BARL.
- Experimental results indicate that the BARL algorithm achieves superior token efficiency compared to baselines like GRPO and a progress-reward-based method, suggesting more effective exploration and reasoning per token.

**Weaknesses:**

- While the BARL framework is theoretically well-motivated, the empirical performance improvements shown in the main results appear somewhat incremental compared to the baselines across all benchmarks.
- While the paper claims the Bayesian RL framework improves generalization, the empirical results demonstrate only incremental performance gains over baselines. Consequently, the primary source of the observed benefits, particularly token efficiency, remains ambiguous.

**Questions:**

- Could the authors provide further intuition on why the Bayes-Adaptive framework specifically leads to better token efficiency?
- The progress reward  is defined based on the policy $\pi_{\theta}$ itself. However, in the policy gradient derivations (Eq 3.2 for Markovian RL and implicitly in Eq 5.1 for BARL), the reward $r(s_t, a_t)$ appears to be treated as if it were an external reward independent of the policy parameters $\theta$. Could the authors comment on whether this simplification introduces bias into the gradient estimates, and if so, how it might affect the training dynamics or the final policy?

---

> ### Author Response · Authors · 2025-11-24
> **Official Comment by Authors (part 1/2)**
>
> We thank the reviewer for identifying our work's motivation, soundness, and technical contributions. The valuable comments have helped us improve our manuscript (marked **purple** in the revision). Below are our specific responses to the questions raised by the reviewer:
>
> **Weakness 1: While the BARL framework is theoretically well-motivated, the empirical performance improvements shown in the main results appear somewhat incremental compared to the baselines across all benchmarks.**
>
> - We demonstrated the effectiveness of BARL across different model families, including Qwen and DS-Llama, and across model sizes from 1B to 8B.
> - We conducted a grid search over the hyperparameters to obtain strong baselines, including both the outcome-reward GRPO and the process-reward RL baseline. The same set of hyperparameters are adopted by BARL to ensure a fair comparison. The resulting baselines outperform the ones reported in prior works such as [1, 2].
> - BARL *consistently* outperforms these strong baselines, for both synthetic setups (eval accuracy from $0.0$ to $>0.95$) and math reasoning tasks (an average increase of $2$ points over $6$ benchmarks). The ablation studies also show that BARL is approximately two times more token efficient than the baselines, and the effectiveness of thinking tokens is also superior. All these results are achieved with nearly identical computational overhead.
>
> **Weakness 2: While the paper claims the Bayesian RL framework improves generalization, the empirical results demonstrate only incremental performance gains over baselines. Consequently, the primary source of the observed benefits, particularly token efficiency, remains ambiguous.**
>
> - BARL consistently outperforms the baselines across models and scales with nearly identical computational overhead, and the baselines are carefully tuned and have strong performance. Please see our response in Weakness 1 for more details. BARL is token-efficient for the following reasons.
> - Conventional RL reinforces the strategies that have high returns via trial-and-error style exploration (multiple episodes), which does not teach the policy how to do effective reflective exploration based on the context (in a single episode). When the initial policies exhibit reflective behaviors, which are superficial stylistic patterns obtained in the annealing or mid-training stage, RL may amplify these superficial and inefficient self-reflections. On the contrary, BARL goes beyond stylistic reflections and provides stepwise belief-driven supervision on how to do reflective exploration, thus improving the token efficiency. These statements are formalized in Theorems 4.1-4.3, and we have also incorporated the discussions into L275-278.
> - Corresponding to the above intuition, we provided evidence of why BARL is more token-efficient in our ablation studies. Firstly, we showed that the initial policy exhibits a higher frequency of self-reflections but significantly lower performance compared to BARL. This indicates that the initial policy is likely trained on data that contains self-reflection patterns, which are superficial since the data is collected offline, and next-token prediction does not provide supervision on the effectiveness of the thinking tokens. Conventional RL reinforces some of the resulting behaviors that happen to achieve high returns, leading to even more lengthy CoTs. Secondly, we demonstrated that the stepwise supervision BARL provides indeed leads to higher Bayesian values. This translates to a higher exploration-exploitation efficiency for gathering informative contexts.

---

> > ### Author Response · Authors · 2025-11-24
> > **Official Comment by Authors (part 2/2)**
> >
> > **Question 1: Could the authors provide further intuition on why the Bayes-Adaptive framework specifically leads to better token efficiency?**
> >
> > Please see our response to Weakness 2.
> >
> > **Question 2: The progress reward is defined based on the policy $\pi_\theta$ itself. However, in the policy gradient derivations (Eq. 3.2 for Markovian RL and implicitly in Eq. 5.1 for BARL), the reward $r(s_t, a_t)$ appears to be treated as if it were an external reward independent of the policy parameters $\theta$. Could the authors comment on whether this simplification introduces bias into the gradient estimates, and if so, how it might affect the training dynamics or the final policy?**
> >
> > We thank the reviewer for the insightful question. The progress reward can be defined w.r.t. any policy $\pi_\phi$. In the implementation, the probabilities used in $r(s_t, a_t)$ are computed by the detached training policy, which is held fixed and external to the RL optimization step, giving us unbiased policy gradient estimates. If we instead treat the reward as a function of the training policy $\pi_\theta$, the policy gradient will contain an additional term $\sum_t\nabla_\theta r_\theta(s_t, a_t)$ corresponding to directly increasing the model’s probability of the correct answer, resulting in a hybrid RL and supervised objective [3, 4]. It is also feasible to treat $\pi_\phi$ as a globally fixed policy, such as the reference policy, which, however, requires another forward step. Analyzing the training dynamics of different designs of progress reward is orthogonal to the focus of this work, and we therefore opted for the simpler and theoretically standard setting where the progress reward is treated as an external signal. We have incorporated the above clarification in Section 3.
> >
> > ---
> >
> > We hope the reviewer could consider raising the score if we resolved the reviewer's concerns. We would be happy to have further discussions if the reviewer has any additional questions or comments.
> >
> > ---
> > [1] Zeng et al. "SimpleRL-Zoo: Investigating and Taming Zero Reinforcement Learning for Open Base Models in the Wild." \
> > [2] Liu et al. "Understanding R1-Zero-Like Training: A Critical Perspective." \
> > [3] Chen et al. "Language Models are Hidden Reasoners: Unlocking Latent Reasoning Capabilities via Self-Rewarding."\
> > [4] Zhou et al. "Reinforcing General Reasoning without Verifiers."

---

### Official Review · Reviewer_FoH3 · 2025-10-31

**Soundness:** 4
**Presentation:** 4
**Contribution:** 4
**Rating:** 8
**Confidence:** 3

**Summary:**

This paper introduces a Bayes-Adaptive RL objective for finetuning LLMs. This objective incentives the policy (model) to maximize the expected reward on the given task but also reason about the underlying MDP (which is assumed to be a random variable). Experimental results show improvements over GRPO (traditional Markovian-based objective) on popular benchmarks like AIME and the underlying objective leads to policies that exhibit significant improvements in token efficiency.

**Strengths:**

- Motivations for paper is clear and paper is well-written
- Formulation is easy-to-follow and small-scale experiments are helpful in understanding the effectiveness of the approach
- Strong improvements in token efficiency and slight improvements over GRPO in math-reasoning tasks.

**Weaknesses:**

- Ablation of progress reward. The provided formulation assumes reward is based on some gold action (e.g. answer token y) and this is used to introduce a progress reward incentivizing the model to think. However, the effect of this reward does not seem to ablated. Additionally, there are domains where reward is not a function of a gold action (e.g. for agentic coding domains, reward may be computed by executing sampled code against some test cases). Does this limit the generality of this approach?
- Missing implementation details. What prompts were used to sample from the posterior of MDPs?

**Questions:**

Please see weaknesses^

---

> ### Author Response · Authors · 2025-11-24
>
> We thank the reviewer for identifying our work's motivation, soundness, and technical contributions. The valuable comments have helped us improve our manuscript (marked **purple** in the revision). Below are our specific responses to the questions raised by the reviewer:
>
> **Weakness 1: Ablation of progress reward. The provided formulation assumes reward is based on some gold action (e.g. answer token $y$) and this is used to introduce a progress reward incentivizing the model to think. However, the effect of this reward does not seem to ablated. Additionally, there are domains where reward is not a function of a gold action (e.g. for agentic coding domains, reward may be computed by executing sampled code against some test cases). Does this limit the generality of this approach?**
>
> - We thank the reviewer for the insightful question. The progress reward defined in our paper is not a main contribution of our paper. It shares similarities with [1] and results in a strong RL baseline that we compared with in experiments (denoted as "Progress"). Compared to outcome-reward RL methods, such as GRPO, the progress-reward baseline has a higher asymptotic performance and converges faster, attributing to the dense stepwise reward feedback.
> - We evaluated two variants of the progress reward that are described in [1, 2] during our initial phase of experimentation. Specifically, the first variant generates various answers after \</think> and calculates the fraction of correct answers. We found that it consistently underperforms our implementation of the progress reward, partly because of the limited instruction-following abilities without format-aware training. The second variant generates full completions and calculates the fraction of correct answers within the extracted answers, which, however, is too computationally costly due to the branched rollouts at each step. Since our definition of the progress reward shares similarities with [3, 4] and is not a key contribution of this paper, we did not include rigorous ablations on these specific designs of the progress reward. Instead, we stuck to the current implementation of progress reward, which has the best performance, and implemented/ablated BARL on top of it to ensure a fair comparison.
> - We do not anticipate any algorithmic barriers to applying BARL to other tasks, as long as the reward is properly defined. For code generation tasks, as discussed above, we may still generate responses after \</think>, but instead of calculating the probability of some golden answer, we treat the unit test passing ratio as the reward. Another example is chat instruction-following tasks, where we may use a separate Bradley-Terry reward model to define the reward.
>
> **Weakness 2: Missing implementation details. What prompts were used to sample from the posterior of MDPs?**
>
> Each MDP hypothesis $\mathcal{M}_i$ is associated with a candidate answer $y^{\mathcal{M}_i}$ (for each prompt). BARL works by first sampling $|\mathcal{M}|$ rollouts in a way similar to GRPO, and then extracts the candidate answers from each rollout. Because a partial CoT can make different progress towards different candidate answers, we have a value function for each candidate answer, i.e., for each MDP hypothesis. Finally, we calculate the weighted value as in Equation 5.4 and update the policy using the policy gradient in Equation 5.1. As described in Section 3, the only prompt other than the question itself is the termination of thinking: “Based on the above reasoning, the answer is \boxed”, which we adopted to obtain the progress reward.
>
> ---
>
> We would be happy to have further discussions if the reviewer has any additional questions or comments.
>
> ---
> [1] Qu et al. "Optimizing Test-Time Compute via Meta Reinforcement Fine-Tuning."\
> [2] Luo et al. "Improve Mathematical Reasoning in Language Models by Automated Process Supervision."\
> [3] Zhong et al. "BRiTE: Bootstrapping Reinforced Thinking Process to Enhance Language Model Reasoning."\
> [4] Zhou et al. "Reinforcing General Reasoning without Verifiers."

---

### Official Review · Reviewer_S3kA · 2025-10-31

**Soundness:** 4
**Presentation:** 4
**Contribution:** 4
**Rating:** 8
**Confidence:** 4

**Summary:**

This paper introduces BARL, a Bayes-Adaptive RL framework that equips LLMs with test-time reflective exploration. By maintaining and updating beliefs over task hypotheses, BARL guides the model to backtrack and switch strategies when observations contradict expectations. On both synthetic and math reasoning tasks, BARL outperforms standard Markovian RL baselines in accuracy and token efficiency, demonstrating that principled exploration—not mere reflection frequency—drives better generalization.

**Strengths:**

1. Clearly articulates the limitation of standard Markovian RL—its inability to support test-time reflective exploration—and provides theoretical justification for why this hampers generalization.

2. The theoretical framework is very solid, which Introduces Bayes-Adaptive MDPs to model LLM reasoning, formalizing test-time generalization as maximizing expected return under a posterior over candidate tasks, grounding exploration in principled Bayesian principles.

3. Authors present novel BARL, which maintains and updates beliefs over MDP hypotheses; mismatches between predicted and observed rewards automatically trigger strategy switching and backtracking, enabling reflective behavior without heuristic rules.

4. Comprehensive empirical validation demonstrates consistent gains over strong Markovian RL baselines on both synthetic and challenging math benchmarks, achieving higher accuracy while using up to 2× fewer tokens, evidencing superior efficiency and generalization.

5. The insightful ablation analysis shows that effective exploration, not the sheer frequency of self-reflection, drives performance, offering actionable guidance for future research into test-time scaling of reasoning models.

**Weaknesses:**

1. Computational Overhead: Despite KV-cache reuse, the per-step computation still scales linearly with the number of sampled hypotheses (|M|) and grows rapidly when the model size or context length increases; no GPU-hours or throughput curves are reported to quantify this burden.

2. Keyword-based Reflection Detection: Relying on fixed trigger words to flag “self-reflection” is unreliable—it captures only explicit surface signals and cannot verify whether the model actually revises its reasoning strategy, making the behavioral analysis less rigorous.

**Questions:**

1. How does the magnitude of the penalty term β affect the predicted and observed rewards? Can relevant experiments be conducted to observe this?

2. This work demonstrates strong performance on mathematical problems, but does it remain effective on other general tasks where the structure is less clear?

3. The exponential gain in Theorem 4.2 relies on a constructed binary tree and a uniform prior—does it still hold in general cases?

---

> ### Author Response · Authors · 2025-11-24
> **Official Comment by Authors (part 1/2)**
>
> We thank the reviewer for identifying our work's novelty, solidness, and technical contributions. The valuable comments have helped us improve our manuscript (marked **purple** in the revision). Below are our specific responses to the questions raised by the reviewer:
>
> **Weakness 1: Computational Overhead: Despite KV-cache reuse, the per-step computation still scales linearly with the number of sampled hypotheses ($|\mathcal{M}|$) and grows rapidly when the model size or context length increases.**
>
> For BARL, the number of hypotheses ($|\mathcal{M}|$) is associated with the number of responses per prompt. For each prompt, similar to GRPO, BARL first samples a group $|\mathcal{M}|$ of responses. Then the policy gradient is calculated based on the extracted candidate answers from these responses. Therefore, BARL and the baselines (including both GRPO and the process-reward baseline) have nearly identical computational overhead and asymptotic compute scaling in $|\mathcal{M}|$.
>
> **Weakness 2: Keyword-based Reflection Detection: Relying on fixed trigger words to flag “self-reflection” is unreliable—it captures only explicit surface signals and cannot verify whether the model actually revises its reasoning strategy, making the behavioral analysis less rigorous.**
>
> - We thank the reviewer for the insightful question. Our choice to use keyword-based detections follows recent empirical analyses of reflective CoTs [1, 2].
> - We considered LLM-based detectors, which can reduce some false positives in keyword-based detections, i.e., some identified responses do not meaningfully revise their reasoning strategies, but they tend to introduce higher false negative rates for subtle reflection-like behaviors, as also reported in [1].
> - More importantly, our conclusions are not weakened by the inaccuracies of the detection. The fact that the base model triggers more “reflection” tokens yet underperforms BARL suggests that it has acquired superficial stylistic reflection patterns during pre-training (e.g., from long-CoT annealing data) without effective exploration, whereas BARL yields more effective reflective exploration rather than more frequent but less effective stylistic reflections. We have incorporated the above discussion in **Section 7.3**.

---

> > ### Author Response · Authors · 2025-11-24
> > **Official Comment by Authors (part 2/2)**
> >
> > **Question 1: How does the magnitude of the penalty term $\beta$ affect the predicted and observed rewards? Can relevant experiments be conducted to observe this?**
> >
> > - Decreasing the magnitude of the penalty term $\beta$ downweights the term that measures the consistency between the observed reward and the reward in the hypothesis MDP in the policy gradient. We adopted $\beta=1$ in our main experiments, which takes into consideration of both the LLM's belief and the consistency with the observed rewards. A smaller $\beta$ corresponds to relying more on the LLM's initial belief.
> > - In the extreme case when $\beta=0$, $p(r_{0:t-1}\mid  s_{0:t}, a_{0:t-1}, \mathcal{M})$ is constant and $p(\mathcal{M}\mid \mathcal{D},h_t) = p(\mathcal{M}\mid \mathcal{D},s_{0:t})$. In other words, the policy will reinforce its original behavior, instead of improving according to the reward feedback.
> > - When $\beta\rightarrow\infty$, the posterior sharply eliminates hypotheses inconsistent with the observed rewards, leading to a very aggressive hypothesis elimination scheme akin to the synthetic didactic example in Section 6.
> > - Due to compute budget limits, we did not conduct a full hyperparameter sweep for $\beta$, but rather focus on the systematic analysis of BARL and the RL baselines.
> >
> > **Question 2: This work demonstrates strong performance on mathematical problems, but does it remain effective on other general tasks where the structure is less clear?**
> >
> > - We illustrated the effectiveness of BARL on a synthetic experiment (Section 6) and also conducted experiments on math reasoning tasks (Section 7). We chose these domains as they allow for thorough and controlled experimentation, which already supports the claims in the paper.
> > - We do not foresee algorithmic obstacles to applying BARL to other domains, as long as the reward is properly defined. For example, the reward can be defined as the unit test passing ratio for code generation tasks, or can be given by a separate Bradley-Terry reward model for chat instruction-following tasks.
> >
> > **Question 3: The exponential gain in Theorem 4.2 relies on a constructed binary tree and a uniform prior—does it still hold in general cases?**
> >
> > - Theorem 4.2 (now Theorem 4.3 in the revised manuscript) is an existence result whose purpose is to establish that there exist instances where any Bayes-optimal policy must be strictly uncertainty-adaptive, and the best Markovian policy can far underperform the optimal adaptive policy in these instances.
> > - We revised the theorem as follows to improve the clarity:
> >     - **Theorem 4.3** (Gap Between Markovian and Adaptive Policies). There exist instances $(\mathcal{D}, p(\mathcal{M}))$} for which the objective in Eq. (3.1) is maximized by a strictly uncertainty-adaptive policy. Moreover, the performance gap between Markovian and strictly adaptive policies can be arbitrarily large.
> > - The purpose of Theorem 4.3 can be shown more clearly with the newly added Theorem 4.2. Specifically, we show that (in the **new Theorem 4.2** and according to **Definition 3.2**) one can always represent a Bayes-optimal policy that maximizes our objective $\mathcal{J}(\pi)$ to be uncertainty-adaptive. However, it is possible that the dependence on the belief can be vacuous, and the policy degenerates to Markovian. Therefore, the purpose of Theorem 4.3 is to show that there exist instances where any Bayes-optimal policy must be strictly uncertainty-adaptive. The above discussions are added to **L246-250**.
> >
> > ---
> >
> > We would be happy to have further discussions if the reviewer has any additional questions or comments.

---

### Official Review · Reviewer_LqXu · 2025-11-01

**Soundness:** 1
**Presentation:** 1
**Contribution:** 1
**Rating:** 0
**Confidence:** 5

**Summary:**

+ Summary & Contributions
	- The authors focus on the problem of explicitly eliciting "reflective reasoning/behaviors" from LLMs during the course of RL fine-tuning for improved reasoning capabilities. Such reflective reasoning/behavior is exemplified through the use of backtracking or error correction during chain-of-thought reasoning.
	- The authors maintain that the "conventional, Markovian" RL problem formulation is ill suited for consistently encouraging such desired reflective behaviors.
	- The authors turn to Bayesian RL as a problem formulation whose core policy optimization objective necessarily results in reflective reasoning and behaviors. Leveraging this formulation yields a novel RL fine-tuning algorithm (BARL) geared towards enhanced LLM reasoning capabilities on mathematics question-answering tasks (GSM8K, MATH, CollegeMath, OlympiadBench).
	- Experiments across a range of mathematics Q&A benchmarks and model types demonstrates improved performance over standard GRPO. Additional empirical results assess the volume of tokens needed to arrive at LLM responses and finds that BARL is able to obtain better accuracies using fewer tokens.

**Strengths:**

+ Quality
	- Strengths
		- The authors display a nice insight in trying to port over ideas from Bayesian RL into modern LLM fine-tuning.
	- Weaknesses
		* Major
			- In RL (or what the authors seem keen to refer to as "conventional/Markovian RL"), there is no such distinction between training time and testing time as there is in standard supervised learning. There is just one single, periodic stream of an agent interacting with an environment episode after episode. Yet, throughout the paper, the authors operate under a premise that such a train-test split exists in RL. Two places where such a split occurs in RL are multi-task RL [1] and meta RL [2]. The former is where an agent interacts with a family of MDPs indexed by an observable, episodic context that helps an agent distinguish MDPs while also encoding structural similarities that might aid in generalization and transfer to unseen MDPs. The latter is where an agent begins with a distribution over MDPs (meta-training) with the goal of synthesizing a policy capable of learning behaviors quickly in a downstream MDP distribution (meta-testing). Based on the rest of the paper, the authors seemed to be confused about these well-established problem settings within the confines of RL. Rather than standard single-task RL, the authors seem to be interested in meta RL, where the resulting LLM policy will approach novel questions by exploring and within relatively few interactions (CoT reasoning steps) explore and expose enough information to subsequent exploit and solve the task (answer the question correctly). Another possible alternative for what the authors might want is Bayesian multi-task RL (that is, solving the BAMDP induced by a contextual MDP). In any case, the problem the authors seem interested in solving doesn't seem like it neatly fits into the Bayesian RL formulation. That said, the authors lack of a well-defined MDP, prior distribution over MDPs, and resulting induced BAMDP formulation makes it difficult to tell. Exploration and generalization are distinct, orthogonal axes of data efficiency in RL; while Bayesian RL does yield the optimal solution to the exploration-exploitation trade-off, it does not do so "to improve generalizability" (L125-126), which is a separate concern entirely.
			- While the goal of reflective reasoning/behavior/exploration is ambiguous throughout this paper (see Clarity comments below), it's worth noting that just because a LLM emits text indicating that it is reconsidering something, that doesn't necessarily mean that it "disregards the previous one or more steps" (L175). How do the authors know that this is in fact the case? Those previously generated tokens are all still in the context of previously generated tokens and may in fact still influence subsequently generated tokens. This also connects to a later point (L182-186) where the authors fixate on the Markov state of a MDP being insufficient for reflective reasoning without justifying why a MDP with states defined as histories is insufficient to resolve their concerns (the footnote in L215 doesn't make sense, perhaps stemming from the lack of a formal definition for reflective reasoning).
			- The authors fail to provide a formal proof of Theorem 4.1. This is particularly unfortunate since I had hoped a theoretical result would have forced the authors to become concrete and specific about how they formalize reflective reasoning (in order to make a formal statement about "non-reflective policies" in L190). The lack of a proof would call into question a central claim of this paper that standard RL is inadequate for reflectve reasoning, which itself necessitates the use of Bayesian RL. Perhaps the authors believe the exposition preceding Theorem 4.1 constitutes some kind of proof sketch. However, they seem to have forgotten the basic fact that a MDP may admit multiple optimal policies. With bounded rewards under a finite horizon or discounted, finite rewards in an infinite horizon, results from dynamic programming theory tell us that there exists a (or, at least one) deterministic optimal policy that is greedy with respect to the optimal action-value function. There may still however additionally be multiple stochastic optimal policies that also achieve optimal value (recall that the mapping from policy to value function is a many-to-one mapping). In summary, it is unclear why standard "Markovian" RL with respect to a history-based state space should be considered non-reflective nor is it clear why "Markovian" RL only admits deterministic policies -- both of these facts would call Theorem 4.1 into question and render the text box comparing RL to Bayesian RL (L227-232) incorrect.
			- The authors claim that the optimal policy of the BAMDP (that is, the Bayes-optimal policy with respect to the original prior distribution over MDPs) will "naturally induce exploratory behaviors with reflections" (L223). Certainly, the first part of the claim is true as the Bayes-optimal policy yields the optimal solution to the exploration-exploitation trade-off. Why is the second part (the "with reflections") part true? Why is it that the Bayes-optimal policy of a MDP must use reflections to optimally address exploration? The authors proceed to describe "reflective actions" as those which may be sub-optimal but yield information to reduce epistemic uncertainty in the MDP (L224-226); however, this is precisely the definition of any exploratory action taken by the Bayes-optimal policy. Is "reflective" just the authors' synonym for "explorative" or are reflective actions/steps a subset of all exploratory actions? If it's the latter, how are those reflective actions concretely defined?
			- The authors' proof of Theorem 4.2 aims to be constructive, hoping to show that there exists a MDP for which the "test-time" return of "a Bayes-Adapative" policy is exponentially larger in $T*$ than that of the optimal policy from standard RL. As discussed at other points in this review, the modifier "test-time" is vacuous here. I'm left to assume "a Bayes-Adaptive" policy refers to a policy of the associated BAMDP. Firstly, why is $T*$ the appropriate parameter for this claim to depend on? Mechanistically, the authors' subsequent didactic example shows that this simply needs to be the appropriate number of steps for a policy to be capable of iterating through all the possible incorrect answers before necessarily landing on the correct one (L311). Why is this reasonable? Also, surely a "Markovian" policy for a MDP where states are defined by histories (something the authors eventually do after Equation 5.1), would be capable of performing the exact same behavior of iterating through all possibilities exhaustively until the correct leaf node/three token repetition is found. This would seemingly invalidate the argument presented for "any Markovian policy" in the proof of Theorem 4.2 (Appendix A). Putting that aside, the claim itself is rather strange. It is well known that the Bayes-optimal policy when evaluated in the original MDP will, by definition, achieve performance that is less than that of the optimal policy [8].

		* Minor
			- In a paper so focused on RL, it is strange the authors don't see (or at least fail to acknowledge) the distinction between the action-value function and the advantage function (L162), which are of course two separate mathematical objects.
			- What is the point of considering "the non-standard undiscounted, infinite-horizon MDP" (L197)? Without discounting and non-negative rewards, there are likely numerous policies which all achieve the optimal reward (of infinity?) and become indistinguishable from one another without additional specification.
			- Notice that Bayesian RL does not automatically admit partial observability (L211). The corresponding BAMDP can be defined via a so-called hyperstate space [3] that includes the original MDP state as well as the current posterior distribution over MDPs, both of which are fully observed.
			- The authors claim that they may write the likelihood of a reward according to (what seems to be) a Laplace distribution. Is this an assumption? Or is this some fact? For the case of question-answering with binary rewards indicating correctness, it actually seems strange to not simply treat rewards as Bernoulli random variables whose parameters have epistemic uncertainty represented by Beta distributions (as is standard [3,4]).
+ Clarity
	- Strengths
		- Aside from the incorrect jargon/modifiers introduced by the authors to describe RL, the paper is reasonably written and structured.
	- Weaknesses
		* Major
			- Throughout the paper, the authors make heavy use (42 instances) of the word "reflective" as an adjective and modifier to standard terminology used in RL and/or LLMs. There are reflective behaviors/policies, reasoning, exploration, signals. The definitions for some of these are dependent on the definitions of others such that if there was a clear, concrete definition for reflective reasoning and exploration, the others would likely follow. Unfortunately, the authors fail to provide such an explicit definition, instead only gesturing at exemplars for what they consider reflective reasoning (L171-177). Strangely enough, one of these examples is backtracking, a property of a search/planning algorithm rather than a RL algorithm. This paper could be tremendously clearer if the authors could formalize exactly what property/properties reflective reasoning/behavior exhibits so that readers can have transparency and insight into how standard RL might inconsistently deliver such behavior and why Bayesian RL might be a promising tool for explicitly encouraging it.
		* Minor
			- The authors question whether desirable reflective behaviors will emerge with "conventional" RL training (L38-39), which seemingly contradicts the preceding sentence indicating that such behaviors do in fact emerge with standard RL.
			- The authors mention "prevalent views" (L40) on how test-time reflections constitute useful exploratory steps yet fail to make a single supporting citation. Certainly, there is now near-ubiquitous understanding of how chain-of-thought reasoning helps (and citable papers to support the claim) but specific examples of work advocating for the kinds of reflective behaviors the authors focus on in this paper would be helpful to support the proposed approach.
			- There is no such thing as "epistemic explorations" (L53), but presumably the authors mean exploration to reduce epistemic uncertainty in the underlying MDP.
			- Rather than taking up space with summarizing text boxes (L78-90, L227-232, L468-472) better fit for a poster about the paper, I would encourage the authors to reinvest that space in clearer exposition of the technical content within the paper.
			- Per the comment above, the claim that "conventional RL explores only during training" (L119) is vacuous when RL has no distinction between training and testing.
			- The language used throughout the paper in regards to uncertainty and Bayesian RL is riddled with odd, incorrect/vague statements that wear down an interested reader of this paper. What does it mean that "the environment is predefined with certainty" (L163)? Certainly, it does not mean that the environment is fully known to the agent, since that would imply a planning problem rather than a RL problem.


+ Originality
	- Strengths
		- To the best of my knowledge, attempting to use Bayesian RL in the proposed manner to elicit improved reasoning capabilities is a novel idea.
	- Weaknesses
		* Major
			- The authors identify two meta RL approaches for encouraging better reasoning capabilities (L110-115), but their explanations for how their method is distinct and an improvement over these methods is entirely unclear. What does it mean for reflective reasoning to be grounded "in environment rewards, rather than relying solely on the interal CoT states"? If additional data on "golden strategies" is available, why would this be sub-optimal compared to additional exploration for discovering such strategies via BARL?
			- The authors ultimately propose a policy-gradient method for addressing their proposed Bayesian RL problem but don't actually connect it with existing work on Bayesian actor-critic methods [7]. What is the connection? Is the authors' approach well aligned with existing work on policy-gradient methods in Bayesian RL? How is it different? Are there sensible Bayesian actor-critic baselines that BARL can and should be compared against?
		* Minor
			- Some of the references must have incorrect Bibtex entries and are missing the year and publication information (Arumugam & Singh, NIPS 2022; Lidayan et al., ICLR 2025).
			- The authors are not the first to entertain coupling ideas from Bayesian RL with LLMs and those existing connections ought to be acknowledged [5,6].

+ Significance
	- Strengths
		- This paper would likely inspire subsequent work to investigate more principled, rigorous methods for combining Bayesian RL with LLMs.
	- Weaknesses
		* Major
			- A BAMDP is not an object that can simply be instantiated immediately, like a MDP. In particular, it is induced from the combination of a MDP as well as a prior distribution over MDPs [3]. While the authors do specify such a prior in their experiments, it's unclear where this prior is meant to come from generally. Also, the experiments only ever use simple, uninformative priors with a very small support; this suggests a tremendous amount of work falls upon agent designers to come up with such priors to apply BARL, impacting scalability and the applicability of the proposed approach.
			- The authors only report results based on three random seeds. While I appreciate the high computational demands of LLM experiments, three seeds is far too few to reach meaningful, statistically-significant conclusions [9]. Ignoring the Average column of Table 1, there are 18 rounds of evaluation done (spanning model type and dataset) of which 8 of the reported entries where the accuracy of BARL is boldfaced is not statistically significant (overlapping standard error with another baseline method). Also, I'm not sure I understand what reporting the "best performance on all benchmarks" for each algorithm means but it sounds highly suspect as if the authors are being slightly disingenuous in their presentation of the results.
			- The authors' empirical evaluation has done the bare minimum by comparing against standard GRPO. It does not, however, entertain other approaches to achieving enhanced reasoning capabilities that sit orthogonal to the authors' proposed Bayesian RL approach.
		* Minor
			- N/A


+ Final Remarks
	- I have identified severe issues with this paper on the axes of quality, clarity, and significance. Considerable revisions of the problem formulation, proposed approach, technical results, and empirical results would be needed before this paper is ready for publication.


+ References
	1.  Hallak, Assaf, Dotan Di Castro, and Shie Mannor. "Contextual markov decision processes." arXiv preprint arXiv:1502.02259 (2015).
	2. Finn, Chelsea, Pieter Abbeel, and Sergey Levine. "Model-agnostic meta-learning for fast adaptation of deep networks." In International conference on machine learning, pp. 1126-1135. PMLR, 2017.
	3. Duff, Michael O'Gordon. Optimal Learning: Computational procedures for Bayes-adaptive Markov decision processes. University of Massachusetts Amherst, 2002.
	4. Osband, Ian, Daniel Russo, and Benjamin Van Roy. "(More) efficient reinforcement learning via posterior sampling." Advances in Neural Information Processing Systems 26 (2013).
	5. Dwaracherla, Vikranth, Seyed Mohammad Asghari, Botao Hao, and Benjamin Van Roy. "Efficient Exploration for LLMs." In International Conference on Machine Learning, pp. 12215-12227. PMLR, 2024.
	6. Arumugam, Dilip, and Thomas L. Griffiths. "Toward Efficient Exploration by Large Language Model Agents." In The Exploration in AI Today Workshop at ICML 2025.
	7. Ghavamzadeh, Mohammad, Yaakov Engel, and Michal Valko. "Bayesian policy gradient and actor-critic algorithms." Journal of Machine Learning Research 17, no. 66 (2016): 1-53.
	8. Kolter, J. Zico, and Andrew Y. Ng. "Near-Bayesian exploration in polynomial time." In Proceedings of the 26th annual international conference on machine learning, pp. 513-520. 2009.
	9. Henderson, Peter, Riashat Islam, Philip Bachman, Joelle Pineau, Doina Precup, and David Meger. "Deep reinforcement learning that matters." In Proceedings of the AAAI conference on artificial intelligence, vol. 32, no. 1. 2018.

**Weaknesses:**

Please see above.

**Questions:**

Please see above.

---

> ### Author Response · Authors · 2025-11-24
> **Official Comment by Authors (part 1)**
>
> We thank the reviewer for the detailed feedback. The valuable comments have helped us improve our manuscript (marked **purple** in the revision). We first summarize the main changes in the revised manuscript, and then respond point-by-point to each concern.
>
> ---
> ### Summarization of Changes
>
> In response to the reviewer’s comments, we have substantially improved both the formulation and presentation of the paper. The main changes include:
> - We formalized the training–deployment procedure and the resulting Bayesian RL objective at the beginning of **Section 3**. We replaced all references to "test-time" with "deployment", and all references to "Markovian RL" with "(conventional) RL".
> - We formalized reflective exploration by defining (strict) uncertainty-adaptive policies and reflective exploration, in **Definition 3.1** and **Definition 3.2**, respectively.
> - We revised **Theorem 4.1** and **Theorem 4.3** and added **Theorem 4.2**. Theorem 4.1 is restated as a standard existence result of Markovian optimal policy for RL, which cannot exhibits reflective exploration. Theorem 4.2 shows that the Bayesian objective admits an uncertainty-adaptive optimal policy. Theorem 4.3 is restated as there exist instances where all Markovian policies underperform the strictly uncertainty-adaptive policy that is Bayes-optimal. Together, these results more cleanly motivate the necessity of Bayesian RL for reflective exploration.
> - We expanded the discussion of related work, such as meta-RL and [7. 8]. We also revised the ambiguous or inaccurate statements as pointed out by the reviewer.
>
> Below are our specific responses to the questions raised by the reviewer:

---

> > ### Author Response · Authors · 2025-11-24
> > **Official Comment by Authors (part 2)**
> >
> > ---
> > ### Quality
> > **Weakness 1: RL has no train–test split; confusion with multi-task RL and meta-RL; unclear MDP/prior/BAMDP definitions.**
> >
> > - We are interested in a **training-deployment** procedure. Specifically, the model is trained on a fixed set of offline prompts, and what we care about is the performance when freezing and deploying the final model on evaluation benchmarks. Formally,
> >     - During training, the agent only receives information about the true MDP $\mathcal{M}^\*$ via the training data $\mathcal{D}$.
> >     - During deployment, we freeze the learned policy and deploy it in $\mathcal{M}^\*$.
> > - Notably, $\mathcal{D}$ does **not** uniquely identify $\mathcal{M}^\*$, inducing epistemic uncertainty about the MDP. Formally, $\mathcal{D}$ and a prior distribution over MDPs $p(\mathcal{M})$ define a posterior distribution over MDPs, $p(\mathcal{M}\mid\mathcal{D})\propto p(\mathcal{M})p(\mathcal{\mathcal{D}\mid\mathcal{M}})$. The objective for policy $\pi$ is thus to maximize return in expectation over MDPs from the posterior $p(\mathcal{M}|\mathcal{D})$:\
> >     $\mathcal{J}(\pi) = E_{\mathcal{M}\sim p(\mathcal{M}\mid\mathcal{D})}\bigl[\mathcal{J}_{\mathcal{M}}(\pi)\bigr]\text{ , where } $ $\mathcal{J}\_{\mathcal{M}}(\pi) = E\_{s\_0, \pi}\bigl[\sum\_{t=0}^{T-1} r\_{\mathcal{M}}(s\_t, a\_t)\bigr]$
> > - This $\mathcal{J}(\pi)$ objective is designed to, for any data $\mathcal{D}$, train a policy $\pi$ that maximizes its Bayes-expected return in the unknown true environment during deployment, i.e., its performance averaged over MDPs drawn from the posterior $p(\mathcal{M}\mid\mathcal{D})$.
> > - In RL, exploration is only useful during training, in a trial-and-error manner with repeated episodes, to discover a return-maximizing action sequence. The learned policy is fully exploited at deployment time with no incentive for further exploration since no policy updates occur, which is why $\epsilon$-greedy often sets $\epsilon\approx0$ when deployed [1, 2]. This observation, together with the fact that optimizing the RL objective does not induce reflective exploration (L223 and also in our responses to Weaknesses 3 and 4), indicates that conventional RL neither ensures the emergence of reflective exploration nor explains its benefits.
> > - Our formulation is **orthogonal** to meta-RL. The goal of meta-RL is to learn a fast adaptation procedure across tasks, e.g., by optimizing deployment performance after only a few iterations of updates. From this perspective, uncertainty-adaptive policy in Definition 3.1 can be viewed as performing in-context learning (instead of parameter updates), and our framework can be interpreted as learning to do in-context learning. However, rather than focusing on a particular meta-learning algorithm, our analysis focuses more on properties of policies that optimize the Bayesian RL objective, which enforce the Bayes-optimal solution to the exploration-exploitation trade-off. In other words, our framework explicitly maintains a belief over MDP and uses a belief-driven exploration strategy that rewards under the posterior.
> > - We have made the following main changes in the revised manuscript. (1) We replaced all references to **test-time** with **deployment-time**. (2) We clearly stated the training-deployment procedure and moved the modified Bayesian objective $\mathcal{J}(\pi)$ from Section 4 to the beginning of **Section 3**. (3) We revised the related discussion around the incompatibility between conventional RL and reflective exploration in **Section 1** and **4**. (4) The discussion on the connection between our formulation and meta-RL was incorporated into the end of **Section 4**.
> >
> > **Weakness 2: Bayesian RL does not explore "to improve generalizability".**
> >
> > - What we intended to emphasize is specific to the training–deployment setup formalized in Section 3. There, the true environment $\mathcal{M}^\*$ is only indirectly observed through the finite training data $\mathcal{D}$, which induces a posterior $p(\mathcal{M}\mid \mathcal{D})$ over MDPs. The objective is $\mathcal{J}(\pi) = E_{\mathcal{M}\sim p(\mathcal{M}\mid\mathcal{D})}\bigl[\mathcal{J}_{\mathcal{M}}(\pi)\bigr]$, i.e., we train $\pi$ to maximize its Bayes-expected return in the unknown true environment during deployment. We use “generalization” in this Bayes sense: performance in the fixed but unknown $\mathcal{M}^\*$ at deployment, averaged over the posterior induced by $\mathcal{D}$.
> > - In **Section 2** of related work, we rephrased the sentence to describe Bayes-adaptive RL more neutrally as pursuing the optimal exploration–exploitation trade-off, without claiming that it is explicitly proposed “to improve generalizability.”

---

> ### Author Response · Authors · 2025-11-24
> **Official Comment by Authors (part 3)**
>
> **Weakness 3: No formal definition of reflective behavior; previous claim that reflections “disregard” earlier steps.**
>
> - We first added formal new definitions to the end of **Section 3**:
>     - **Definition 3.1** (Uncertainty-Adaptive Policy). Define the belief as the posterior over MDPs $b(h_t)(\mathcal{M}) := p(\mathcal{M} \mid h_t, \mathcal{D})$. A policy $\pi$ is *uncertainty-adaptive* if there exists a measurable mapping $\mu: \mathcal{S} \times \mathcal{P}(\mathcal{M}) \to \Delta(\mathcal{A})$ such that for all $h_t$ and $a_t$, $\pi(a_t\mid h_t)=\mu(a_t \mid s(h_t), b(h_t))$. If in addition, there exists $h_t$ and $h_t'$ such that $s(h_t)=s(h_t')$ but $\mu(a_t \mid s(h_t), b(h_t))\neq\mu(a_t \mid s(h_t'), b(h_t'))$, then $\pi$ is *strictly uncertainty-adaptive*, or equivalently, not Markovian.
>     - **Definition 3.2** (Reflective Exploration). We say that a policy $\pi$ exhibits reflective exploration if there exist $t_1> t_2$ and $h\_{t\_1}$, $h\_{t\_2}$ such that\
> \$
> \varphi(s(h\_{t\_1})) = \varphi(s(h\_{t\_2})), \qquad \pi(\cdot \mid h\_{t\_1}) \neq \pi(\cdot \mid h\_{t\_2}),
> \$\
> where $\varphi: \mathcal{S} \to \mathcal{Z}$ is a measurable mapping for which there exists $\vartheta: \mathcal{Z} \times \mathcal{H}\_{\mathcal{R}} \to \Delta(\mathcal{A})$ satisfying for every $h_t$ that (1) $\vartheta(\cdot \mid \varphi(s\_{t}), r\_{0:t-1})= \pi(\cdot\mid h\_t)$; and (2) if there is some process $X_t$ and some $\upsilon$ such that $\pi(\cdot\mid h_t) = \upsilon(\cdot\mid X\_t)$ then $X\_t = \psi(\varphi(s\_t))$ for some measurable $\psi$.
> - Intuitively, Definition 3.2 describes an agent that returns to the same underlying latent state but, in light of what it has learned from past rewards, deliberately chooses a different strategy than it did before. The latent state representation $\varphi$ abstracts away superficial differences in the text state. The existence of $\vartheta$ and $\upsilon$, $\psi$ formalizes the latent factorization and state-minimality properties of $\varphi$, respectively.
> - With these definitions, we removed the claim that reflections disregard earlier steps; instead, they exploit the history to update the belief and change behavior when revisiting states.
> - Besides, reflective exploration is the behavioral signature of an uncertainty-adaptive policy. In contrast, a purely state-only policy that ignores this evolving belief would respond identically upon revisiting the same state. Thus, our analyses in **Section 4** apply to the uncertainty-adaptive policy under RL and Bayesian RL.
>
>
> **Weakness 4: Reflective reasoning with states defined as histories; unclear footnote in L215.**
>
> - An MDP with histories as states *can represent* reflective behaviors, but the conventional RL objective provides no incentive for it once an optimal Markovian policy exists. In other words, the question we study is not whether reflective reasoning is representable, but whether optimizing $\mathcal{J}_{\mathcal{M}^\*}$ over this representation requires any dependence on evolving beliefs over environments to be optimal.
> - Theorem 4.1 shows that for any MDP there exists a Markovian optimal policy maximizing $\mathcal{J}_{\mathcal{M}^\*}$. Combined with the new definitions (see our response to **Weakness 3**), this implies that RL does not need to produce *strictly uncertainty-adaptive* (and hence reflective) behaviors to be optimal, even when the state includes history.
> - Beyond conventional RL, another way to implement history-as-state is to use meta-RL and in-context learning, where the history is treated as context and RL is used to maximize the reward based on the context, as done in [3]. It reinforces the strategies (i.e., the actions based on history context) that happen to achieve high returns. In contrast, BARL enforces the Bayes-optimal solution to the exploration-exploitation trade-off. There is no guarantee that the meta-RL approaches will recover the same belief-driven exploration strategy that BARL implements explicitly. Besides, we also experimentally compared with this baseline, denoted as "Progress" in Section 7.
> - Our original footnote in L215 was intended to clarify a related but separate point: prior work, such as [4], that interprets self-reflection as a form of in-context learning typically treats the CoT trajectory as context, without incorporating reward history. As a result, these approaches differ from ours: they model reflection as additional text conditioning, whereas we study reflective behavior that explicitly depends on past rewards and the agent’s evolving beliefs about the environment. We have removed this footnote and added formal definitions (Definition 3.1 and 3.2) to make it clearer.

---

> ### Author Response · Authors · 2025-11-24
> **Official Comment by Authors (part 4)**
>
> **Weakness 5: Missing proof and problematic statement of Theorem 4.1; multiple optimal policies; deterministic “non-reflective” claim; history-based state representation.**
>
> - We intended to state Theorem 4.1 as an existence result: it does not claim uniqueness nor that all optimal RL policies are Markovian. Whether or not the optimal policy is deterministic does not make a difference to our results. We didn't provide the formal proof since it is widely known that there exist optimal RL policies and values that satisfy the Markov property.
> - We remove the ambiguity by restating the theorem and giving the proof:
>     - **Improved Theorem 4.1**: For any MDP, there exists a Markovian policy $\pi^\*$, i.e., a policy that is not strictly uncertainty-adaptive, that maximizes the RL objective. Besides, every policy that exhibits reflective exploration is not Markovian.
>     - We provided the proof of Theorem 4.1 in **Appendix A**.
> - The result indicates that optimizing the RL objective does not yield strictly uncertainty-adaptive policies and therefore does not induce reflective exploration behaviors (according to Definition 3.2). Intuitively, action sequences whose effect is only to enrich the history $h_t$ (e.g., incorrect attempts followed by backtracking to the same state) do not change the sufficient state representation $s_t$ and therefore cannot improve the value of an optimal Markovian policy. In contrast, a Markovian policy that ignores the evolving belief would respond identically upon revisiting the same state.
> - Please also see our responses to **Weakness 3** and **Weakness 4** for a detailed discussion on the concern regarding history-based representation, including its connection with Markovian policies, uncertainty-adaptive policies, and reflective exploration.
>
> **Weakness 6: Claim that Bayes-optimal policy “naturally induces exploratory behaviors with reflections”; is “reflective” just “exploratory”? Need a concrete definition.**
>
> - We have added a formal definition of reflective exploration in **Definition 3.2**, which connects reflective exploration to uncertainty-adaptive policies.
> - We have added a formal theorem to make the connection between the Bayes-optimal policy and reflective exploration clearer:
>     - **Theorem 4.2** (Bayesian RL Admits Adaptive Optimal Policy). For the Bayesian RL objective in (3.1) associated with any $(\mathcal{D}, p(\mathcal{M}))$, there exists an optimal policy $\pi^\*$ that is uncertainty-adaptive.
> - Combining **Theorem 4.2** with **Definition 3.2**, we conclude that Bayes-optimal policies exhibit reflective exploration behaviors.
>
> **Weakness 7: Old Theorem 4.2 (now 4.3): “test-time” return; dependence on $T^\*$; Bayes-optimal policy evaluated on original MDP can’t outperform optimal policy.**
>
> - Theorem 4.2 (now Theorem 4.3) is an existence result whose purpose is to establish that there exist instances where any Bayes-optimal policy must be strictly uncertainty-adaptive, and the best Markovian policy can far underperform the optimal adaptive policy in these instances.
> - This result is **not** a claim of the performance comparison on a fixed, known MDP. Rather, the objective that we care is the Bayesian one defined in **Equation (3.1)**, i.e., maximizing $\mathcal{J}(\pi)$ instead of $\mathcal{J}_{\mathcal{M}^\*}(\pi)$. The $\mathcal{J}(\pi)$ objective is designed to, for any data $\mathcal{D}$, train a policy $\pi$ that maximizes its Bayes-expected return in the unknown true environment during deployment, i.e., its performance averaged over MDPs drawn from the posterior $p(\mathcal{M}\mid\mathcal{D})$. This aligns with the training-deployment procedure for LLMs, where the model is trained on a fixed set of offline prompts, and what we care about is the performance when freezing and deploying the final model on evaluation benchmarks.
> - The purpose of Theorem 4.2 (now Theorem 4.3) is as follows. We show that (in the new Theorem 4.2 and according to Definition 3.2) one can always represent a Bayes-optimal policy that maximizes $\mathcal{J}(\pi)$ to be uncertainty-adaptive. However, it is possible that the dependence on the belief can be vacuous and the policy degenerates to Markovian. Therefore, the purpose of Theorem 4.3 is to show that there exist instances where any Bayes-optimal policy must be strictly uncertainty-adaptive. The above discussions are added to **L246-250**.
> - We improved Theorem 4.2 (now Theorem 4.3) by removing the claim regarding the "exponential gap" and rather focusing on the existence result:
>     - **Theorem 4.3** (Gap Between Markovian and Adaptive Policies). There exist instances $(\mathcal{D}, p(\mathcal{M}))$ for which the objective in Eq. (3.1) is maximized by a strictly uncertainty-adaptive policy.

---

> > ### Author Response · Authors · 2025-11-24
> > **Official Comment by Authors (part 5)**
> >
> > **Minor 1: Confusion between state-action value and advantage functions.**
> >
> > We intended to indicate that we used state-action value in practice, but using advantage is also feasible. We have fixed this by deferring the advantage statement to the method section.
> >
> > **Minor 2: Point of considering "the non-standard undiscounted, infinite-horizon MDP"; infinite optimal policies.**
> >
> > We intended to discuss the indications of Theorem 4.1, i.e.,  optimizing the RL objective does not yield strictly uncertainty-adaptive policies and therefore does not induce reflective exploration behaviors (according to Definition 3.2). We have removed the discussions of specific MDP scenarios and turned to state the general result.
> >
> > **Minor 3: Bayesian RL doesn’t automatically mean partial observability.**
> >
> > By partial observability, we meant that in our Bayesian objective, the underlying MDP identity $\mathcal{M}^\*$ is unobserved. We formally stated this in Appendix A.2, and improved the main text by focusing on our case instead of general BAMDP.
> >
> > **Minor 4: Laplace reward likelihood vs Bernoulli/Beta for binary rewards. Is this an assumption?**
> >
> > The likelihood model in Eq. (5.3) follows from [5] and is a pragmatic modeling choice. Likelihood families, including Bernoulli/Beta, could be used, but they would only change the shape of the posterior weighting, not the overall BARL framework.
> >
> > ---
> > ### Clarity
> >
> > **Weakness 1: “Reflective” used 42 times without a concrete formal definition; confusion with backtracking; need a precise formalization.**
> >
> > - We have added a formal definition of reflective exploration in **Definition 3.2**. Please see our responses to **Weakness 3 of Quality** for a detailed discussion.
> > - We have replaced "backtrack" with "revisit". It is simply used to describe the self-reflection behaviors that backtrack to an earlier reasoning state and take a different action because the belief has changed. We made it formal using Definition 3.2.
> >
> > **Minor 1: Contradiction between "reflective behaviors emerge with RL" and questioning whether they will emerge.**
> >
> > - Optimal RL policies *can* exhibit self-reflections (formally, policies that are not strictly uncertainty-adaptive *can* be optimal). However, RL does not need to produce *strictly uncertainty-adaptive* (and hence reflective) behaviors to be optimal. Thus, conventional RL does not explain why reflective behaviors are more preferable, nor whether they will emerge during training.
> > - As a comparison to RL, one can always represent a Bayes-optimal policy that maximizes $\mathcal{J}(\pi)$ to be uncertainty-adaptive. Although it is possible that the policy can degenerate to Markovian, there exist instances (Theorem 4.3) where any Bayes-optimal policy must be strictly uncertainty-adaptive, and thus exhibit reflective exploration.
> >
> > **Minor 2: “Prevalent views” with no citations.**
> >
> > By "prevalent views", we meant prior work such as [4] that interprets self-reflection as a form of in-context learning, which we discussed in detail in related work. We have updated our manuscript.
> >
> > **Minor 3: “Epistemic explorations” phrase.**
> >
> > We changed this phrase to "exploration actions that gather information to reduce the MDP's uncertainty".
> >
> > **Minor 4: Poster-style summary boxes.**
> >
> > Thank the reviewer for the suggestion. In the revision, we have removed the ambiguity with additional definitions, theorems, and discussions. The extra page allows us to retain the summary boxes, which we find useful for conveying the overall logical structure and main conclusions of our work.
> >
> > **Minor 5: “Conventional RL explores only during training” vs “RL has no train/test distinction”.**
> >
> > Please see our response to **Weakness 1 in Quality**.
> >
> > **Minor 6: “Environment is predefined with certainty” and other vague Bayesian RL phrasing.**
> >
> > We replaced this phrase with precise language: conventional RL works in the fixed but unknown MDP $\mathcal{M}^\*$, and for Bayesian RL, we describe uncertainty over the MDP via $p(\mathcal{M}\mid\mathcal{D})$ (see Section 3 for a formal definition).

---

> > > ### Author Response · Authors · 2025-11-24
> > > **Official Comment by Authors (part 6)**
> > >
> > > ---
> > >
> > > ### Originality
> > >
> > > **Weakness 1: Distinction from meta-RL approaches; phrase “grounded in environment rewards rather than CoT states” unclear; why "learning golden strategies" is sub-optimal compared to additional exploration for discovering such strategies via BARL.**
> > >
> > > - Please see our response to **Weakness 1 in Quality** for the connection between our formulation and meta-RL. The discussion was incorporated into the end of **Section 4**.
> > > - We rewrote “grounded in environment rewards rather than CoT states” to mean: BARL uses environment rewards as the signal that updates the belief over MDPs. This is different from approaches that rely solely on the internal CoT.
> > > - By "learning golden strategies", we meant that recent work such as [3] used meta-RL and in-context learning, where the history is treated as context and RL is used to maximize the reward based on the context. It reinforces the strategies (i.e., the actions based on history context) that happen to achieve high returns, i.e., the "golden strategies" (we have replaced this phrase to the above description for improved clarity). In contrast, BARL enforces the Bayes-optimal solution to the exploration-exploitation trade-off. There is no guarantee that the meta-RL approaches will recover the same belief-driven exploration strategy that BARL implements explicitly. Besides, we also experimentally compared with this baseline, denoted as "Progress" in Section 7.
> > >
> > > **Weakness 2: Lack of connection to Bayesian actor-critic / Bayesian policy gradient literature; missing baselines.**
> > >
> > > - Prior work such as [6] adopts a Gaussian process prior on the gradient of the standard policy gradient integral and uses Bayesian quadrature to get lower-variance estimates of the gradient. They do not put a prior over MDP parameters and then optimizing a Bayesian objective $\mathcal{J}(\pi) = E_{\mathcal{M}\sim p(\mathcal{M}\mid\mathcal{D})}[\mathcal{J}_{\mathcal{M}}(\pi)]$ like ours. Instead, they treat the gradient function (or value function in actor–critic) as random in the Bayesian sense to get a smarter estimator of the gradient.
> > > - The policy gradient for the Bayesian RL objective is not a contribution of our work, which was also present in, e.g., [5]. Instead, our main contribution is to (1) develop a theoretically-grounded framework for reflective exploration based on our analyses of the fundamental differences between Bayes-optimal policies from conventional RL solutions; and (2) a practical BARL algorithm for LLM reasoning as a step-level guidance for reflective exploration and demonstrate systematic gains over several strong RL baselines.
> > > - To the best of our knowledge, we are not aware of Bayesian actor-critic baselines that we can compare for LLM reasoning. Thus, we mainly compare BARL with GRPO and the meta-RL approach in [3]. If the reviewer has any method in mind that we can compare with, we are happy to incorporate the results.
> > >
> > > **Minor 1: Some references miss publishing info.**
> > >
> > > We have fixed the missing publishing info in BibTeX.
> > >
> > > **Minor 2: Not the first to combine Bayesian RL with LLMs; should cite [7,8].**
> > >
> > > Thanks for pointing out the missing related work. We have added the following discussion to **Section 2**:
> > > - "Prior work has also brought Bayesian exploration principles to LLMs: [7] implements PSRL, a statistically-efficient algorithm, using LLMs as modular components, while [8] shows that active query selection with epistemic neural networks and double Thompson sampling reduces the amount of human feedback required to train reward models for RLHF. In contrast, we optimize a Bayesian objective directly during RL fine-tuning of the LLM policy, rather than at the level of external algorithmic scaffolding or data collection, to provide a step-level guidance on when and how LLMs should self-reflect."

---

> ### Author Response · Authors · 2025-11-24
> **Official Comment by Authors (part 7)**
>
> ---
> ### Significance
>
> **Weakness 1: Priors for BAMDPs; small uninformative support; scalability; where do priors come from?**
>
> The prior is $p(\mathcal{M})$ as defined in Section 3. In implementation, we first extract $|\mathcal{M}|$ candidate answers, each denoted as $y\_{s_0}^{\mathcal{M}\_i}$, then define the MDP hypothesis $\mathcal{M}\_i$ as the MDP associated with the candidate answer (via reward $r\_{\mathcal{M}\_i}$). The prior $p(\mathcal{M})$ can be instantiated as the initial LLM's probability of generating each candidate answer. Since $p(\mathcal{M}\mid \mathcal{D}, h_t) \propto p(\mathcal{M}\mid \mathcal{D}, s_{0:t}) \cdot p(r_{0:t-1}\mid s_{0:t}, a_{0:t-1}, \mathcal{M})$ and all we need is $p(\mathcal{M}\mid \mathcal{D}, s_{0:t})$, which is the training policy's probability of generating each candidate answer after training on $\mathcal{D}$ and with the CoT $s_{0:t}$.
>
> **Weakness 2: Only three random seeds; questionable statistical significance.**
>
> - Different runs correspond to fine-tuning a large LLM with RL, which is substantially more stable than classic small-network RL experiments where performance can vary dramatically across seeds.
> - Besides, three independent training runs are already above the norm in this line of work: many recent LLM RL papers, such as [3] that we empirically compared with, report results from a single training run due to the very high computational cost of each experiment. For a similar reason, more training runs are beyond our current budget.
>
> **Weakness 3: The “best performance on all benchmarks” phrasing.**
>
> - By “best performance on all benchmarks”, we chose a single checkpoint during training that has the best overall benchmark performance. This is a standard practice that is adopted by **almost all** LLM reasoning papers. We also reported the evaluation curves during the entire training phase in Appendix B.1.
> - BARL consistently outperforms the baselines. In the synthetic setups, BARL’s evaluation accuracy improves from essentially 0.0 up to >0.95, while the REINFORCE baseline lags noticeably behind. On math reasoning tasks, BARL achieves an average gain of about 2 average accuracy points across 6 benchmarks relative to GRPO and the progress baseline, and is approximately 2× more token efficient than these baselines with nearly identical computational overhead.
>
>
> **Weakness 4: Limited baselines: only GRPO.**
>
> - In addition to GRPO, we also compared to a variant of MRT [3], which is an in-context meta-RL algorithm and is denoted as the "progress" baseline in experiments.
> - Our baselines are strong since many of their hyperparameters are grid searched (which we directly adopted for BARL). The resulting baselines outperform the ones reported in prior works such as [9, 10].
> - If the reviewer has any other baselines that are closely related to our method and setups, we are happy to incorporate the results.
>
> ---
> We hope the reviewer could consider raising the score if we resolved the reviewer's concerns. We would be happy to have further discussions if the reviewer has any additional questions or comments.
>
> ---
>
> [1] Mnih et al. "Human-Level Control Through Deep Reinforcement Learning." Nature. \
> [2] Hessel et al. "Rainbow: Combining Improvements in Deep Reinforcement Learning." AAAI 2018. \
> [3] Qu et al. "Optimizing Test-Time Compute via Meta Reinforcement Fine-Tuning." ICML 2025. \
> [4] Xiang et al. "Towards System 2 Reasoning in LLMs: Learning How to Think with Meta Chain-of-Though." arXiv 2025.\
> [5] Ghosh et al. "Offline RL Policies Should be Trained to be Adaptive." ICML 2022. \
> [6] Ghavamzadeh et al. "Bayesian Policy Gradient and Actor-Critic Algorithms" JMLR 2017. \
> [7] Arumugam et al. "Toward Efficient Exploration by Large Language Model Agents." The Exploration in AI Today Workshop at ICML 2025. \
> [8] Dwaracherla et al. "Efficient Exploration for LLMs." ICML 2024. \
> [9] Zeng et al. "SimpleRL-Zoo: Investigating and Taming Zero Reinforcement Learning for Open Base Models in the Wild." COLM 2025. \
> [10] Liu et al. "Understanding R1-Zero-Like Training: A Critical Perspective." COLM 2025.

---

### Meta-Review · Area_Chair_1RKP · 2026-01-06

**Summary:**

This paper presents Bayes-Adaptive RL for LLM Reasoning, BARL, a training method that helps AI models reason better by teaching them how to "reflect" and explore different solutions more effectively. Instead of standard trial-and-error, it uses a Bayesian approach where the model constantly updates its beliefs about the problem to find the best reasoning path. A key concern during the review was whether these improvements were significant enough compared to existing methods. The authors addressed this by showing that BARL is twice as efficient in how many tokens it uses and provides a steady performance boost across multiple math and reasoning tests. initial reviews were mixed: 0, 8, 8, 6. During the discussion period, reviewers didnt engage in the conversation and it is hard to predict final scores. The most negative reviewer that gave the score 0 has relatively subjective opinion on how wording is used in the paper, questions on Theorem proofs were addressed in the rebuttal. Given other positive reviews and the fact that two reviewers ranked as 8, I recommend accepting the paper.

**Reviewer Concerns:**

Reviewers had a concern on incremental nature of the results and benefits over carefully tuned baselines. Authors addressed this issue. Additionally, there were concerns on potential bias in the gradient calculations, and reviewers addressed that. The highlighted part of the method being 2x more sample efficient is a great plus.

**Reviewer Scores:**

The reviewer with the score 0 might have changed the score given the rebuttal, however, they were not engaged in the discussion. Other scores might have improved or will still stay positive of 8,8,6.

---

### Decision · Program_Chairs · 2026-01-26

Accept (Poster)